# Evaluation of optical particulate matter sensors under realistic conditions of strong and mild urban pollution

Adnan Masic[1], Dzevad Bibic[1], Boran Pikula[1], Almir Blazevic[1], Jasna Huremovic[2], and Sabina Zero[2]

[1]Faculty of Mechanical Engineering, University of Sarajevo, 71000 Sarajevo, Bosnia-Herzegovina
[2]Faculty of Science, University of Sarajevo, 71000 Sarajevo, Bosnia-Herzegovina

**Correspondence:** Adnan Masic (masic@mef.unsa.ba)

**Abstract.** In this paper we evaluate characteristics of three optical particulate matter sensors/sizers (OPS): high-end spectrometer 11-D (Grimm, Germany), low-cost sensor OPC-N2 (Alphasense, United Kingdom) and in-house developed MAQS (Mobile Air Quality System) which is based on another low-cost sensor – PMS5003 (Plantower, China), under realistic conditions of strong and mild urban pollution. Results were compared against a reference gravimetric system, based on Gemini (Dadolab, Italy), $2.3\,\mathrm{m^3/h}$ air sampler, with two channels (simultaneously measuring $PM_{2.5}$ and $PM_{10}$ concentrations). The measurements were performed in Sarajevo, the capital of Bosnia-Herzegovina, from December 2019 until May 2020. This interval is divided into period 1 - strong pollution and period 2 - mild pollution. The city of Sarajevo is one of the most polluted cities in Europe in terms of particulate matter: the average concentration of $PM_{2.5}$ during the period 1 was $83\,\mu\mathrm{g/m^3}$, with daily average values exceeding $500\,\mu\mathrm{g/m^3}$. During period 2, the average concentration of $PM_{2.5}$ was $20\,\mu\mathrm{g/m^3}$. These conditions represent a good opportunity to test optical devices against reference instrument in a wide range of ambient particulate matter (PM) concentrations. The effect of an in-house developed diffusion dryer for 11-D is discussed as well. In order to analyze the mass distribution of particles, a scanning mobility particle sizer (SMPS), which together with the 11-D spectrometer gives the full spectrum from nanoparticles of diameter $10\,\mathrm{nm}$ to coarse particles of diameter $35\,\mu\mathrm{m}$, was used. All tested devices showed excellent correlation with the reference instrument in period 1, with $R^2$ values between 0.90 and 0.99 for daily average PM concentrations. However, in period 2, where the range of concentrations was much narrower, $R^2$ values decreased significantly, to values from 0.28 to 0.92. We have also included results of a 13.5 month long-term comparison of our MAQS sensor with a nearby beta attenuation monitor (BAM) 1020 (Met One Instruments, USA) operated by the United States Environmental Protection Agency (US EPA), which showed similar correlation and no observable change of performance over time.

## 1 Introduction

Analysis of particulate matter represents a key element for the studies of air pollution. Various studies shed light on their effect on health (Downward et al., 2018) and climate (Zhao et al., 2019). In many cases particulate matter is a dominant pollutant among other components of pollution. Therefore, developing a strategy for reliable quantification of particulate matter in ambient air is necessary. The traditional and most accurate approach to measuring the particulate matter concentration in the air is the reference method, based on gravimetric measurements, after the collection of particulate matter by air samplers.

The typical time resolution of such measurements is 24 hours. Although there are portable air samplers, these measurements are usually performed at fixed locations, such as research supersites. Reference systems are expensive and require a lot of laboratory work. Results are not immediately available, because of the time-consuming process of filter treatment. Taking that into account, various governmental institutions usually opt for more affordable and easier to use and maintain – equivalent methods. These are usually fixed, semi-automatic stations equipped with beta attenuation monitors (BAMs). The typical time

resolution of such stations is 1 hour. If maintained and calibrated properly, the equivalent methods should achieve an acceptable level of agreement with the reference. For example, one long-term comprehensive study (Hafkenscheid and Vonk, 2014) performed at 14 different locations across Netherlands, showed that a linear correction $y = 0.91x - 1.6$, applied on raw readings from BAM, was necessary to achieve the requirements of the Guide to the Demonstration of Equivalence (ECWG, 2010).

    Newer methods, based on optical particle sensors (OPS), are nowadays increasingly more popular, particularly low-cost

variants (Zheng et al., 2019; Mukherjee et al., 2019; Tanzer et al., 2019; Morawska et al., 2018). Their typical time resolution is between 1 s and 1 min, and because of their price and size, they can be used in networks to provide better spatial coverage (Martin et al., 2019; Li et al., 2019). Furthermore, they provide information about multiple mass fractions of particulate matter simultaneously, unlike the concentration of single fraction in gravimetric system or BAM. However, there are concerns about their suitability for measuring mass concentrations of ambient PM, since there is a significant measurement uncertainty arising

from the principles of their operation.

    Most commercially available OPS use the Mie scattering theory (Mie, 1908) to determine the size and number of particles within the unit volume of air. The Mie theory provides the solution of the Maxwell equations for the scattering of plane waves on spherical particles. The Mie solution is rather complex, but in order to illustrate the non-linearity of the theory, it will suffice to consider the case where particles are much smaller than the wavelength (of light, since a red laser is commonly used in

practice). In that case the intensity of scattered radiation is given by

$$I = I_0 \frac{1 + \cos^2\theta}{2R^2} \left(\frac{2\pi}{\lambda}\right)^4 \left(\frac{d}{2}\right)^6 \left|\frac{m^2 - 1}{m^2 + 2}\right|^2, \tag{1}$$

where $I_0$ is intensity of the incident radiation, $\theta$ is the scattering angle, $R$ is the distance between the particle and the observing point, $\lambda$ is the wavelength, $d$ is the particle diameter and $m$ is the refractive index of the particle. Thus, in order to calculate the diameter of the particle by measuring the intensity of the scattered radiation, one must assume a value for the refractive index

of the particle. If the particle absorbs nothing from incoming radiation, its refractive index will be real, otherwise it is written in the form

$$m = n + i\kappa, \tag{2}$$

where $\kappa$ is called the extinction coefficient and is related to the absorption coefficient $\alpha$:

$$\alpha = \frac{4\pi\kappa}{\lambda}. \tag{3}$$

Once the size distribution is calculated accross $K$ channels (bins), the total mass concentration of particles will be

$$c_m = \sum_{i=1}^{K} w_i \rho_i V_i N_i, \tag{4}$$

where $V_i$ is the (average) volume, $\rho_i$ is the density of the particles, $N_i$ is the number of particles per unit volume and $w_i$ is the weighting factor for channel $i$. Here we have another cause of OPS uncertainty: the density of particles must be assumed. Regarding the weighting factors, sensor manufacturers calculate values to correct for certain effects, such as the fact that OPS

cannot detect particles which are too small.

Laboratory tests and calibrations of OPS are performed under controlled conditions with known particles, such as polystyrene latex spheres (Walser et al., 2017; Bezantakos et al., 2018), continuously changing monodisperse particles (Kuula et al., 2017, 2020) or multi-modal particles (Cai et al., 2019). Burning chamber is used in some investigations as well (Wang et al., 2015). However, equation (1) is strongly non-linear in terms of refractive index, and in most practical cases corrections for different

particles' optical properties are impossible to implement. Furthermore, densities appearing in equation (4) are not known *a priori*. That explains why it is difficult to calibrate OPS for realistic ambient PM concentration measurements: any laboratory calibration may or may not be applicable to the changing outdoor conditions (Tryner et al., 2020; Crilley et al., 2020).

For outdoor applications, there is an additional problem: hygroscopic growth of particles (Jayaratne et al., 2018; Granados-Muñoz et al., 2015; Di Antonio et al., 2018), which leads to overshoots of OPS if the ambient air humidity is (too) high. An

obvious solution is to dry the air. However, any proper drying system would cost more than many models of OPS and it is rarely seen in combination with low-cost sensors. Analytical corrections are often used: humidity sensors are used to measure the relative humidity of ambient air and some analytical model, like Kohler's theory (Castarède and Thomson, 2018) or Hänel equation (Hänel, 1976), is applied. Later in this paper we will make some observations on this issue.

Due to all the above-mentioned factors, it is always interesting to check how OPS perform in different realistic scenarios.

Numerous papers deal with laboratory calibrations and outdoor evaluations of OPS (Karagulian et al., 2019; Borghi et al., 2018; Chatzidiakou et al., 2019; Magi et al., 2020; Sousan et al., 2016b; Malings et al., 2020; Kelly et al., 2017; Sayahi et al., 2019; Crilley et al., 2018; Zheng et al., 2018; Tasic et al., 2012; Cavaliere et al., 2018; Mukherjee et al., 2017; Sousan et al., 2016a; Zhang et al., 2018; Holstius et al., 2014; Badura et al., 2018). Reported results vary depending on the composition of particulate matter pollution, range of concentrations and meteorological factors. In (Mukherjee et al., 2017) OPC-N2, PMS7003 and 11-R

were compared against BAM-1020 during 12 weeks in the Cuyama Valley, California, USA. Grimm 11-R performed well, while both OPC-N2 and PMS7003 (which is a miniaturized version of PMS5003) produced mediocre performance with heavy low bias. PurpleAir (PMS5003) was tested in (Tryner et al., 2020) using laboratory and field tests. High bias of PMS5003 was observed. In (Magi et al., 2020) PurpleAir (PMS5003) was analyzed for 16 months in Charlotte, North Carolina, USA against BAM-1022, and high bias of PMS5003 that increases with humidity was reported. High mean bias of PurpleAir (PMS5003)

was reported in (Kosmopoulos et al., 2020) as well.

The novelty of this research is a unique combination of instruments and conditions of extremely high urban pollution. The city of Sarajevo is situated in a valley and is affected by strong temperature inversions that appear typically 150 m-300 m above ground level with a very strong temperature gradient in the inversion layer, exceeding 30 K/km (Masic et al., 2019). The inversion episodes were present during most of January 2020. As a consequence, the average monthly concentration of

$PM_{2.5}$ was very high: 167.3 $\mu$g/m$^3$. In contrast to that, monthly average values for March and April 2020 were 21.6 $\mu$g/m$^3$ and 19.6 $\mu$g/m$^3$, respectively. This presented an excellent opportunity to test the performance of OPS in very different pollution

levels. Simultaneously with OPS and reference gravimetric measurements, an SMPS was employed to detect nanoparticles. It can detect particles with diameters from 10 nm up to 1 $\mu$m. While SMPS can count very small particles, 11-D can count larger particles, from 0.25 $\mu$m to 35 $\mu$m in diameter. When they work simultaneously, they can detect (almost) the full range of particles' diameters, with a span of more than three orders of magnitude. This will give detailed insights on the mass distribution of particles.

## 2 Methodology and experimental setup

The experimental facility was located at the Faculty of Mechanical Engineering in the central part of Sarajevo valley (N 43.85424, E 18.39607, 540 m above sea level) and represents well the overall conditions in the city. The reference instrument for measurements of PM concentrations was a Dadolab Gemini air sampler (Figure 1). It is a single device with two completely independent channels ($PM_{2.5}$ and $PM_{10}$ in this campaign). The filter preparation and gravimetric analysis are performed in separate laboratory of Faculty of Science, Department of Chemistry. The air sampler, gravimetric laboratory and all filter procedures satisfied requirements of the standard EN 12341:2014. According to requirements of the standard, all filters were conditioned at relative humidity between 45% and 50%, and temperature between 19 and 21 $^0$C.

Grimm 11-D is a high-end optical particle sizer, with sophisticated construction and ability to count individual particles from 250 nm to 35 $\mu$m in 31 equidistant (on logarithmic scale) channels. It uses a proprietary algorithm and the manufacturer does not share information about the refractive index, density or weighting factors. It was factory calibrated, and equipped with firmware version 12.50. Data was recorded in 1 minute intervals (6 seconds is also possible). Since we use the common term OPS occasionally, it should be noted that 11-D belongs to different category of devices (in comparison to low-cost sensors).

Alphasense OPC-N2 belongs to the category of low-cost sensors. The manufacturer transparently shared most specifications. It has a much simpler construction than 11-D: instead of regulated pump, air flow is provided by 25 mm fan. The device has 16 channels, from 380 nm to 17 $\mu$m. Firmware version 18.2 was used. Refractive index was $n = 1.50 + i0$ and density was 1.65 g/cm$^3$. All other parameters, including weighting factors, were used as firmware default values.

The Plantower PMS5003 could be termed a very low-cost sensor, since its price is lower by an order of magnitude than that of the OPC-N2. Limited specifications don't reveal all operating parameters. From the specification sheet we can conclude that the device uses Mie scattering theory, with detection limit of 300 nm, and has 6 channels. It uses red semiconductor laser, photodetector at $90^0$ scattering angle (Kuula et al., 2020) and 32-bit processor (Cypress CY8C4245, 48 MHz). According to (Tanzer et al., 2019) PMS5003 is a nephelometer, not the particle counter. Air flow is provided by a 20 mm fan. The PMS5003 has two data outputs, one is called SM (standard material, $CF = 1$), and another AE (atmospheric environment). The latter mode is used in our work, since the manufacturer recommends AE mode for ambient air measurements, without further explanation. Figure 2 shows results from our laboratory test using the incense scents as the source of PM. Based on

these results, the relationships between SM and AE modes are

$$
SM_{PM2.5} = \begin{cases} AE_{PM2.5} \text{ for } AE_{PM2.5} \leq 30 \\ \text{nonlinear for } 30 < AE_{PM2.5} \leq 50 \\ 1.5 \times AE_{PM2.5} \text{ for } AE_{PM2.5} > 50 \end{cases}
$$
$$
\text{(5)}
$$
$$
SM_{PM10} = \begin{cases} AE_{PM10} \text{ for } AE_{PM10} \leq 43 \\ \text{nonlinear for } 43 < AE_{PM10} \leq 77 \\ 1.5 \times AE_{PM10} \text{ for } AE_{PM10} > 77 \end{cases}
$$

Based on PMS5003, we have designed MAQS (Mobile Air Quality System) smart sensor. Essentially, it is a modular platform for PMS5003, with options for additional sensors (pressure, temperature, humidity, carbon dioxide, wind speed), GNSS receiver, flash memory, Wi-Fi module and 3D-printed enclosure. Eight MAQS sensors were made and tested prior to the main campaign in order to evaluate consistency between units. Figure 3 shows the results of preliminary outdoor measurements for a batch of 8 MAQS sensors. They showed very good consistency: the coefficient of determination, $R^2$, between any two

sensors from the batch was greater than 0.99 and average readings from all sensors are within $\pm 10\%$ from the average value of the batch of sensors. Data was recorded every minute on a local SD card and remote cloud server simultaneously. Recording interval can be as short as 1 second, but there was no need for that.

    Grimm 11-D and Alphasense OPC-N2 could not be used outdoors without shelter, while MAQS has a special case which provides basic protection for outdoor use. Furthermore, netbook PC was used to record data from the OPC-N2. Outdoor shelter

had to be constructed to accommodate 11-D with power supply, OPC-N2 with PC and SPI adapter, and MAQS (for better protection). Stevenson screen like wooden structure was designed for that purpose. Another MAQS sensor was used at a remote location, for reasons that will be explained later.

    For low-cost sensors (OPC-N2 and MAQS) there was no air dryer or heater, since they are typically used in such conditions. We have designed and constructed diffusion dryer for application on 11-D, which consists of porous stainless steel tube

surrounded with 1 kg of silica gel. The dryer is compact, 25 cm in length with 8 cm external diameter and does not reduce the mobility of the instrument. It was installed only during the period of mild pollution.

    Meteorological parameters were measured using Vantage Pro2 (Davis Instruments, USA) weather station with recording intervals of 15 minutes.

    SMPS is a complex system which consists of a condensation particle counter (CPC), a differential mobility analyzer (DMA)

and a charge conditioner (often inadequately called "neutralizer"). Depending on the characteristics of the DMA, the SMPS can be configured for certain span of particle diameters. We have used Grimm 5.416 high-end SMPS with long DMA which is able to separate particles from 10 nm to 1000 nm in 129 channels, equidistant on a logarithmic scale. Despite the fact that particles with diameter below 10 nm play an important role in nucleation and growth studies (Tiszenkel et al., 2019), their contribution to the mass budget is negligible. Another (larger) in-house developed diffusion dryer was installed at the inlet of

SMPS. A soft X-ray device was used as the charge conditioner. Scanning mode (alternating upscan and downscan) was used for all measurements. One scan takes about 4 minutes (8 minutes for both upscan and downscan). When working parallelly,

SMPS and 11-D form a powerful wide-range spectrometer, which covers a range of particle diameters from 10 nm to 35 $\mu$m in 160 channels. Additionally, there is an overlapping area between 250 nm and 1000 nm where we can see how well these two instruments match. The complex SMPS system was kept indoors (an unavoidable necessity, since both X-ray charger and DMA use very high operating voltages). The air was sampled from outside using a conductive tube of shortest possible length, to avoid particle losses. It was running continuously, except for the periods of maintenance.

A rigorous data validation procedure was used. All instruments were inspected periodically and data logs were analyzed thoroughly. When calculating daily average values, complete and consistent data series were required.

## 3 Results and discussion

During this campaign 296 filters were used in the reference air sampler. After the removal of several blank filters used for periodic verification and those with incomplete sampling (pneumatic system of air sampler failed to load new filters automatically couple of times), 288 filters remained: 143 PM$_{2.5}$ and 145 PM$_{10}$ samples. Figure 2 shows PM$_{2.5}$ and PM$_{10}$ daily average concentrations, together with hourly and daily values of ambient air temperature and relative humidity.

Some modifications of the shelter for 11-D and OPC-N2 were necessary, making those instruments unavailable periodically during December and January. Additionally, more frequent maintenance, such as cleaning of 11-D, was needed when working in extreme conditions. The same stands for SMPS, which was maintained according to the recommendations of the manufacturer. Taking into account difficult operating conditions, the amount of data collected is satisfactory during the period of strong pollution and excellent during the period of mild pollution. The lower limit of detection (LLoD) of PM$_{2.5}$ concentration for evaluated optical aerosol devices is estimated based on their actual field performance. Standard deviation ($\sigma$) was calculated for periods with near-zero ambient PM concentration and average value of $3\sigma$ is estimated LLoD. For PMS5003 our final estimation is 5 $\mu$g/m$^3$. The same value is an estimation of (Magi et al., 2020), calculated by averaging segmented regressions and (Bulot et al., 2019) by combining results from several previous studies. This method applied on OPC-N2 yields LLoD of 2 $\mu$g/m$^3$ and 1 $\mu$g/m$^3$ for 11-D. For reference gravimetric system LLoD was calculated using the blank filters, which were treated exactly the same way as real samples (except the sampling of particulate matter), and the calculated value of LLoD is 0.7 $\mu$g/m$^3$. All measurements below LLoD were discarded during the quality assurance phase.

### 3.1 Strong urban pollution

During the period of strong urban pollution (12/2/2019–3/12/2020), the average value of PM$_{2.5}$ concentration was 82.9 $\mu$g/m$^3$, with minimum daily average value 1.3 $\mu$g/m$^3$ and maximum value 504.9 $\mu$g/m$^3$. In the same period, the average PM$_{10}$ concentration was 95.5 $\mu$g/m$^3$, with minimum value 3.6 $\mu$g/m$^3$ and maximum value 549.0 $\mu$g/m$^3$. The ratio of average values of concentrations PM$_{2.5}$/PM$_{10}$ was 0.87. Very good correlations were observed for all three OPS against the reference instrument (Figure 5). Such a range of ambient PM concentrations was favorable for achievement of high $R^2$ values, but non-linear effects of low-cost sensors were observed too.

Grimm 11-D produced results with $R^2$ values 0.988 and 0.985 for $PM_{2.5}$ and $PM_{10}$ concentrations, respectively. Absolute values were larger than the reference, on average 17.6% for $PM_{2.5}$ and 25.5% for $PM_{10}$. The average ratio $PM_{2.5}/PM_{10}$ measured by 11-D was 0.93. Mean absolute error (MAE) was 13.4 $\mu$g/m$^3$ for $PM_{2.5}$ and 10.8 $\mu$g/m$^3$ for $PM_{10}$. Alphasense OPC-N2 undershoots with respect to the reference values, on average 31.0% for $PM_{2.5}$ and 36.8% for $PM_{10}$, but $R^2$ coefficients are relatively high: 0.903 and 0.920 for $PM_{2.5}$ and $PM_{10}$ respectively. The OPC-N2 measured the ratio $PM_{2.5}/PM_{10}$ to be 0.97. MAE for this sensor was 29.4 $\mu$g/m$^3$ for $PM_{2.5}$ and 34.8 $\mu$g/m$^3$ for $PM_{10}$. MAQS sensor produced surprisingly good $R^2$ values of 0.975 for $PM_{2.5}$ and 0.950 for $PM_{10}$. In terms of absolute values, it overshoots by 31.9% for $PM_{2.5}$ and 49.3% for $PM_{10}$ (on average). The calculated ratio $PM_{2.5}/PM_{10}$ was 0.76. MAE was 35.9 $\mu$g/m$^3$ for $PM_{2.5}$ and 55.2 $\mu$g/m$^3$ for $PM_{10}$. It seems that the Plantower PMS5003 can not accurately determine the $PM_{10}$ fraction. One possible explanation is provided by a laboratory test of PMS5003, where it was found that its size bin [2.5 $\mu$m-10 $\mu$m] is noisy and inaccurate (Kuula et al., 2020). Further investigation of this behavior would be useful.

None of the tested OPS were equipped with an air dryer, and this certainly contributes to overprediction. Yet, Alphasense OPC-N2 with default firmware settings underpredicts values, despite the particle hygroscopic growth effect.

## 3.2 Mild urban pollution

The correlation coefficients changed dramatically in the period of mild pollution (3/13/2020–5/4/2020), as Figure 6 shows. The much narrower range of particulate matter concentrations plays an important role, and even the reference method is less accurate, since the mass difference of loaded and blank filters becomes very small (smaller than 1 mg for 24 h sampling period if PM concentration is below 18 $\mu$g/m$^3$). The average concentration of $PM_{2.5}$ was 19.7 $\mu$g/m$^3$ with a minimum daily average value of 7.1 $\mu$g/m$^3$ and a maximum value of 39.3 $\mu$g/m$^3$. During this period, the average value of $PM_{10}$ concentration was 24.2 $\mu$g/m$^3$, with minimum and maximum values of 7.6 $\mu$g/m$^3$ and 48.8 $\mu$g/m$^3$, respectively. The ratio $PM_{2.5}/PM_{10}$ was 0.81 on average.

This time Grimm 11-D was equipped with a dryer, whose effects will be discussed in the next subsection. The device produced relatively high $R^2$ values 0.868 for $PM_{2.5}$ and 0.917 for $PM_{10}$. The absolute readings underestimated concentrations of $PM_{2.5}$ by 16.3% on average, while $PM_{10}$ were underestimated by 10.9% on average. The $PM_{2.5}/PM_{10}$ ratio was 0.87. This test clearly shows that 11-D is a completely different class of instrument (in comparison to low-cost sensors). When equipped with dryer, 11-D shows level of performance comparable to BAM, at least those reported by Hafkenscheid and Vonk (2014). MAE was 3.0 $\mu$g/m$^3$ for $PM_{2.5}$ and 4.1 $\mu$g/m$^3$ for $PM_{10}$.

Alphasense OPC-N2 did not perform well during the period of mild pollution. Coefficients of determination, $R^2$, were only 0.284 for $PM_{2.5}$ and 0.525 for $PM_{10}$. Absolute readings are worrying: the OPC-N2 underpredicted $PM_{2.5}$ by 67.6% and $PM_{10}$ by 71.6% on average. The ratio $PM_{2.5}/PM_{10}$ was 0.73. MAE was 13.8 $\mu$g/m$^3$ for $PM_{2.5}$ and 15.8 $\mu$g/m$^3$ for $PM_{10}$.

MAQS sensor demonstrated mediocre performance, with $R^2$ values of 0.730 for $PM_{2.5}$ and 0.718 for $PM_{10}$. On average, this sensor overpredicted $PM_{2.5}$ by 30.5% and $PM_{10}$ by 32.6%. The $PM_{2.5}/PM_{10}$ ratio was 0.83, very close to the reference value (contrary to the performance of the sensor in the period of strong pollution). MAE was 7.1 $\mu$g/m$^3$ for $PM_{2.5}$ and 8.2 $\mu$g/m$^3$ for $PM_{10}$.

It would be interesting to test low-cost sensors with a proper dryer as well, but that combination is rarely seen in practice.

## 3.3 Humidity influence

One of the important factors in ambient measurements of PM concentrations is humidity, since the particles reflect more light
(i.e. appear larger) during measurements due to hygroscopic growth. This can be described using Hänel equation:

$$f_\zeta(RH) = \left( \frac{1 - RH}{1 - RH_{\text{ref}}} \right)^{-\gamma}, \tag{6}$$

where $f_\zeta$ is enhancement factor for particle property $\zeta$. Here $RH$ represents the relative humidity and $RH_{\text{ref}}$ is a reference
relative humidity:

$$f_\zeta(RH) = \frac{\zeta(RH)}{\zeta(RH_{\text{ref}})}. \tag{7}$$

It is important to note that the coefficient $\gamma$, which is an indicator of the hygroscopicity of particles, depends on the type of
particles (and changes whenever composition of ambient particles is changed).

If we compare results produced by 11-D relative to the reference, during period 1 (without dryer) and period 2 (with dryer),
we can see that readings of the 11-D were reduced by more than 30%. However, we can not conclude whether it was the
effect of the dryer or the consequence of significantly different ambient conditions. Unfortunately, we have only one 11-D,
so we couldn't measure simultaneously with and without dryer (that's the reason why we used the instrument with the dryer
only in one of the two periods). If we take into account two intervals with similar ambient conditions, with and without dryer
we get following values: from 2/27/2020 to 3/12/2020 average ambient concentration of $PM_{2.5}$ was 21.1 $\mu g/m^3$ while 11-D
(without dryer) measured 21.5% more. In the second interval, from 3/13/2020 to 4/1/2020, ambient concentration was similar,
21.0 $\mu g/m^3$ while 11-D (with dryer) measured 1.4% smaller value. This comparison indicates that the effect of the dryer could
be around 23%. A similar analysis for $PM_{10}$ concentrations gives an estimate of about 20% for the dryer effect.

Grimm 11-D has a very useful feature: internal temperature and humidity sensor. Figure 7 shows self-heating and diffusion
dryer effect on 11-D, by comparing internal and external measurements of temperature and humidity. The average ambient
air temperature from 2/27/2020 to 4/1/2020 was 7.02 $^0$C while the average 11-D internal temperature was 14.27 $^0$C, which
shows a significant difference of 7.25 $^0$C. This self-heating effect reduces internal humidity significantly, and we can see that it
rarely goes beyond 50%. Once the dryer is installed, internal relative humidity is further reduced: the average value of internal
humidity without dryer (2/27/2020-3/12/2020) was 36.2% and with dryer (3/13/2020-4/1/2020) was 21.8% (the ambient air
humidity also dropped in the later period, but nevertheless the effect of the dryer is evident).

After roughly a month, the dryer's performance degraded and the silica gel needed a regeneration (but it wasn't performed
since we didn't want to interrupt measurements when the end of the campaign was near).

Figure 8 shows the long-term (13.5 months) comparison of MAQS and BAM-1020 with time resolution of 1 hour, together
with measured values of ambient air humidity. By averaging all this data we can estimate the influence of humidity on the
MAQS sensor: if we sort the measurements by humidity, subset of points where humidity is below 50% has average bias of
14.3%, for humidity range 50%-70%, bias is 16.5%, for humidity range 70%-85% bias is 31.6% and for humidity rang 85%-

100% bias is 37.3%. If we subtract bias of least humidity subset from bias of highest humidity subset, we can estimate that humidity influence adds up to 23% on PM2.5 readings from MAQS sensor, which is similar result to the analysis of humidity influence on our 11-D with dryer installed. While this influence can not be neglected, it is still relatively modest. Possible reason is the chemical composition of PM without too much hygroscopic components, but that requires a different type of analysis. Relatively modest humidity influence on PMS5003 was also reported in (Jayaratne et al., 2018) and surprisingly low influence is reported by (Kosmopoulos et al., 2020).

## 3.4 SMPS data and wide-range spectrometer

The wide-range spectrometer (SMPS+11D) produced very valuable results. Figure 9 shows the continuous concentration and mass distributions. It is created from hourly average measurements from SMPS and 11-D. A relative density of 1.65 was applied in SMPS software (based on LabVIEW) for the mass calculation. No other corrections were performed and all settings were factory defaults. Selected histograms (hourly average values) are shown in Figure 10. What we can see from figures 9 and 10 is that in the period of strong pollution the dominant mass contribution comes from particles with diameters around 300 nm. In terms of concentrations, particles around 100 nm appear in greatest numbers, with occasional secondary peaks coming from even smaller particles.

In the period of mild pollution, however, we can see that particles larger than 2.5 $\mu$m often appear on histograms (usually about 3 $\mu$m in diameter). Number concentrations still have peaks about 100 nm, but sometimes the distribution is different in favor of even smaller particles, as Figure 10 shows. Again, the largest mass contribution comes from particles around 300 nm.

In the overlapping area, SMPS and 11-D matched very well, almost perfectly for concentrations. Their match was not as good for mass calculations, but that is understandable, taking into account all the factors explained in section 1. Overall, the combination of SMPS and 11-D worked very well and gave the full spectrum of particles, both for number concentrations and mass distribution.

The obtained mass distribution of particles, especially during the period of strong pollution, rises the question on suitability of OPS for measurements of mass concentrations, and resolving different fractions, since they cannot detect small particles that significantly contribute to the total mass. For example, the Alphasense OPC-N2 has a detection limit of 380 nm, and can't detect the particles around 300 nm which form the dominant contribution to the mass budget. The Grimm 11-D, with a detection limit of 250 nm, has a far better potential to resolve mass fractions.

## 3.5 OPS histograms and Aralkum Desert dust

All tested OPS have data bins, with different number of channels, as described in section 2. Figure 11 shows histograms that compare data bins from 11-D, OPC-N2 and MAQS on 1/18/2020 (strong pollution) and 4/16/2020 (mild pollution). It should be noted that we compare here data bins from devices with different specifications and category. As expected, 11-D has ability to count particles below 300 nm, which appear in greatest numbers. Counting efficiency of OPC-N2 is investigated in laboratory conditions using PSL particles in (Sousan et al., 2016a), and the results were good for particles larger than 0.8 $\mu$m while for particles with diameter of 0.5 $\mu$m OPC-N2 the device showed lower detection efficiency (detection limit of OPC-N2 is 0.38

$\mu$m). In our realistic scenario, dominant contribution to the mass comes from particles much smaller than 0.8 $\mu$m (Figures 9 and 10) which is not favorable to OPC-N2.

Contrary to OPC-N2, PMS5003 has problems with coarse particle, as indicated in laboratory test (Kuula et al., 2020). If the fraction of coarse particles is small and steady, PMS5003 performs much better. Ambient conditions in Bosnia-Herzegovina are such most of the time, since the primary source of PM is combustion of coal and biomass. That could explain why PMS5003 performs better than OPC-N2 most of the time. However, different conditions were observed on 3/27/2020 when the dust from Aralkum Desert covered part of Europe, including our test location. During this episode, OPC-N2 performed much better than PMS5003, which wasn't able to determine large fraction of coarse particles correctly (Figure 11). Similar observation about PMS5003 was reported by (Kosmopoulos et al., 2020), when Sahara dust covered Greece.

## 3.6 Long-term performance

Another question about OPS, especially low-cost types, is the drift of performance over time. The PMS5003 sensor uses a semiconductor laser (diode laser) which has a limited lifetime. We have some long-term comparisons of the MAQS sensor with MetOne BAM-1020 operated at a nearby location by the US EPA. Strictly speaking, their station is not collocated with our equipment, but for the distance of only 300 m it is reasonable to assume that the air composition is very similar at these two points, since they are located in the same neighborhood. In order to verify that assumption, we have installed another MAQS sensor at the location of Faculty of Electrical Engineering, University of Sarajevo, which is in immediate vicinity of the US EPA site, and at the same distance from us (about 300 m). Figure 8 shows long-term comparisons of MAQS sensor and BAM-1020, and additional verification of correlation between readings of two MAQS sensors, which was very high ($R^2 = 0.970$, MAE $= 4.7 \, \mu$g/m$^3$ for hourly average values and $R^2 = 0.994$, MAE $= 2.9 \, \mu$g/m$^3$ for daily average values) confirming our assumption that these two locations share the same air, in terms of PM concentrations and properties.

Based on 13.5 months of continuous comparison of MAQS and BAM-1020, hourly average values give $R^2$ coefficient 0.919 and MAE 16.7 $\mu$g/m$^3$. Daily average values produce $R^2$ coefficient 0.980 and MAE 12.2 $\mu$g/m$^3$, while the monthly average values give $R^2 = 0.998$ and MAE $= 11.4 \, \mu$g/m$^3$ (Figure 8).

This leads us to the conclusion that time averaging reduces a lot the influence of variation of PM composition and meteorological variations. If we use a longer time average period, we lose one of the major advantages of low-cost sensors (time resolution), but it is a more natural approach to correcting readings compared to using artificial algorithms like neural networks (Badura et al., 2019) or machine learning (Si et al., 2020). An excellent viewpoint of this issue is given by Hagler et al. (2018). The calibration of a larger number of low-cost sensors can be simplified if they show similar relative performance (to each other) in laboratory and field (Sousan et al., 2018). Floating corrections, even physically justifiable interventions, such as the instantaneous correction for humidity growth of particles, insert a lot of noise, and the benefit is questionable. Depending on ambient conditions, self-heating of the sensor and some other factors, relative humidity may not be accurately determined. Even if we have a very accurate humidity measurement, the hygroscopic growth coefficient will change whenever the composition of PM changes, inevitably injecting noise into the results.

We can also see strong non-linear effects at very high concentrations, above $500\,\mu g/m^3$. In that case a quadratic regression fit will be more suitable.

    During this period of 13.5 months of continuous outdoor operation, the MAQS sensor worked flawlessly without performance drifts. Designed enclosure sufficiently protected the sensor outdoors, while not obstructing air sampling.

## 4   Conclusions

A comprehensive experimental study was carried out with the aim of evaluating the performance of three very different OPS: high-end Grimm 11-D, low-cost Alphasense OPC-N2 and in-house developed MAQS sensor, which is based on another low-cost sensor, the Plantower PMS5003 sensor. The study was performed in realistic conditions of strong and mild urban pollution. The reference instrument was a dual-channel air sampler with gravimetric analysis in separate laboratory. In total 288 filters were collected from 12/2/2019 to 5/4/2020.

During the period of strong urban pollution all three instruments produced very high $R^2$ values. However, during the period of mild urban pollution, these correlation factors dropped significantly, especially for Alphasense OPC-N2 sensor measuring $PM_{2.5}$ parameter. The OPC-N2 underestimated the mass concentrations badly, especially during the period of mild pollution. MAQS sensor overshoots $PM_{2.5}$ concentrations by approximately 30% on average, which is partially caused by hygroscopic growth.

The wide-range spectrometer, which consists of SMPS and 11-D, produced valuable information about distribution of particles, both in number and mass concentrations. Particles with diameters around 100 nm (and sometimes below) represent the dominant fraction in pure numbers, while particles with diameter of around 300 nm give the highest contribution to mass. In the period of mild pollution, particles larger then 2.5 $\mu$m gave a larger contribution than in the period of strong pollution.

    Grimm 11-D performed well in all conditions, and when equipped with dryer, it performed at a comparable level to the
beta attenuation monitor. For the calibration of low-cost sensors, especially those based on PMS5003, we propose a linear or quadratic correction (in case of high pollution levels) with steady coefficients, since the instantaneous corrections insert noise into results.

    Future measurements should further investigate characteristics of OPS in different ambient conditions, influence of humidity and effect of micro-dryers specifically designed for low-cost sensors and mass distributions by means of wide-range spectrom-
eter.

*Data availability.* The underlying datasets for this publication are available at https://doi.org/10.5281/zenodo.3897379

    Furthermore, data from the BAM measurements by US EPA are availble at https://cfpub.epa.gov/airnow

*Author contributions.* AM participated in all phases of this research and wrote the manuscript with contributions from all co-authors. DB and BP performed field work together with AM. AB designed, manufactured and analysed the performance of two diffusion dryers and evaluated the influence of humidity on the readings of 11-D. SZ and JH performed gravimetric measurements, including preconditioning and treatment of the filters, ensuring strict fulfilment of the standard EN 12341:2014

*Competing interests.* The authors declare that they have no conflict of interest.

*Acknowledgements.* We would like to thank to the Embassy of Sweden in Bosnia-Herzegovina for supporting this research and United States Environmental Protection Agency for sharing data publicly.

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

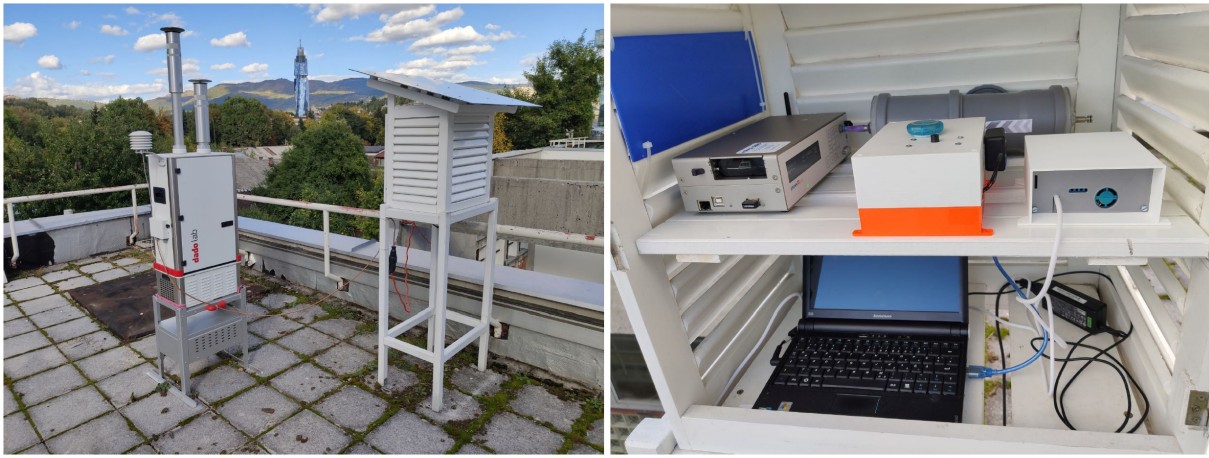

(a) Air sampler and Stevenson screen.

(b) Devices under test.

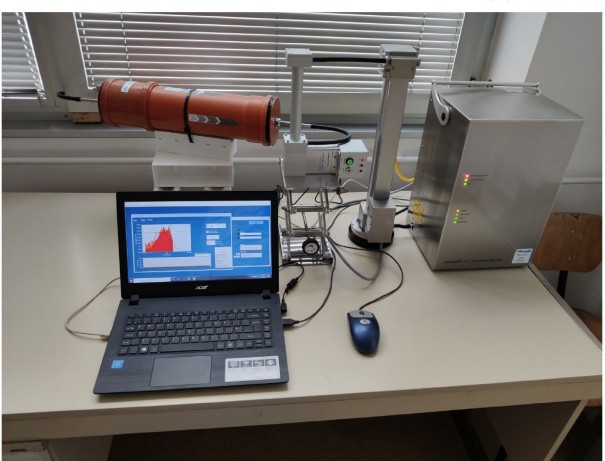

(c) SMPS with dryer.

**Figure 1.** Experimental setup: a) colocated air sampler and Stevenson screen, b) devices under test inside of Stevenson screen: 11-D with dryer, OPC-N2 with SPI adapter and (white-orange) enclosure, MAQS (white enclosure with grey front panel), and c) indoors SMPS with dryer.

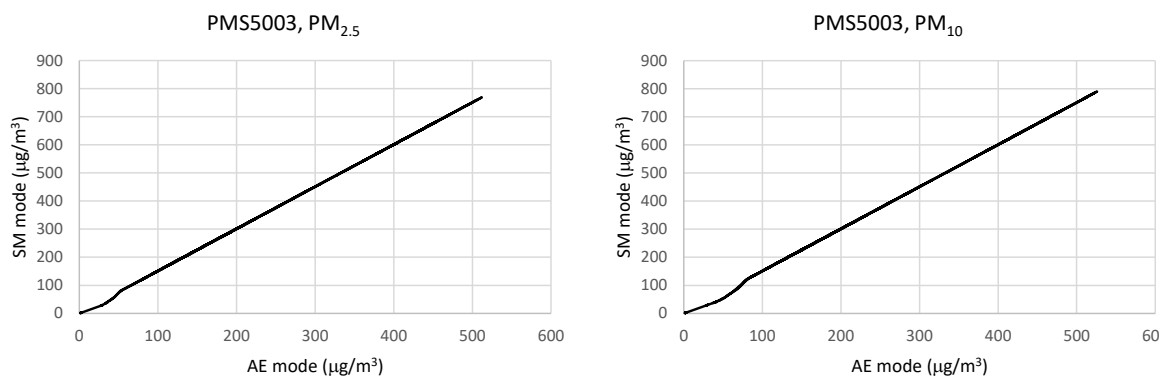

**Figure 2.** AE and SM modes of PMS5003 sensor.

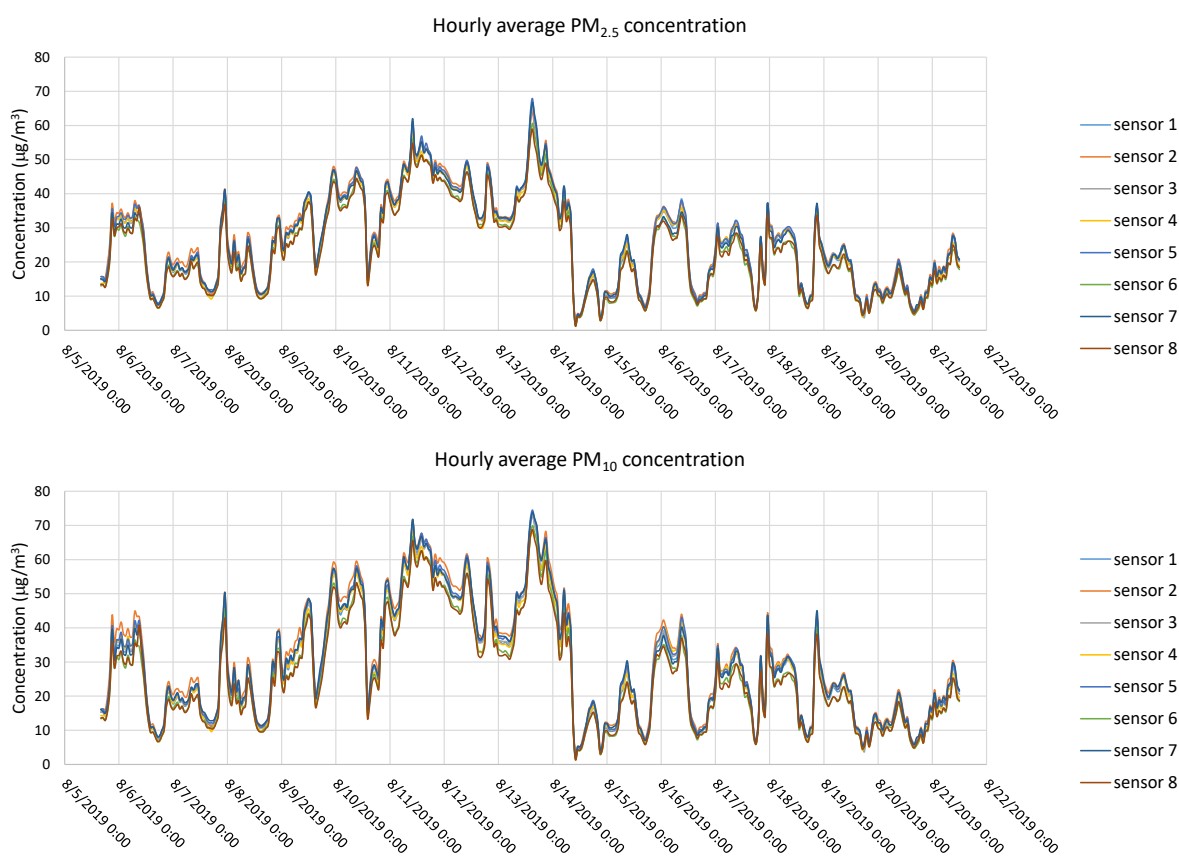

**Figure 3.** Preliminary test of 8 MAQS sensors, outdoor measurements.

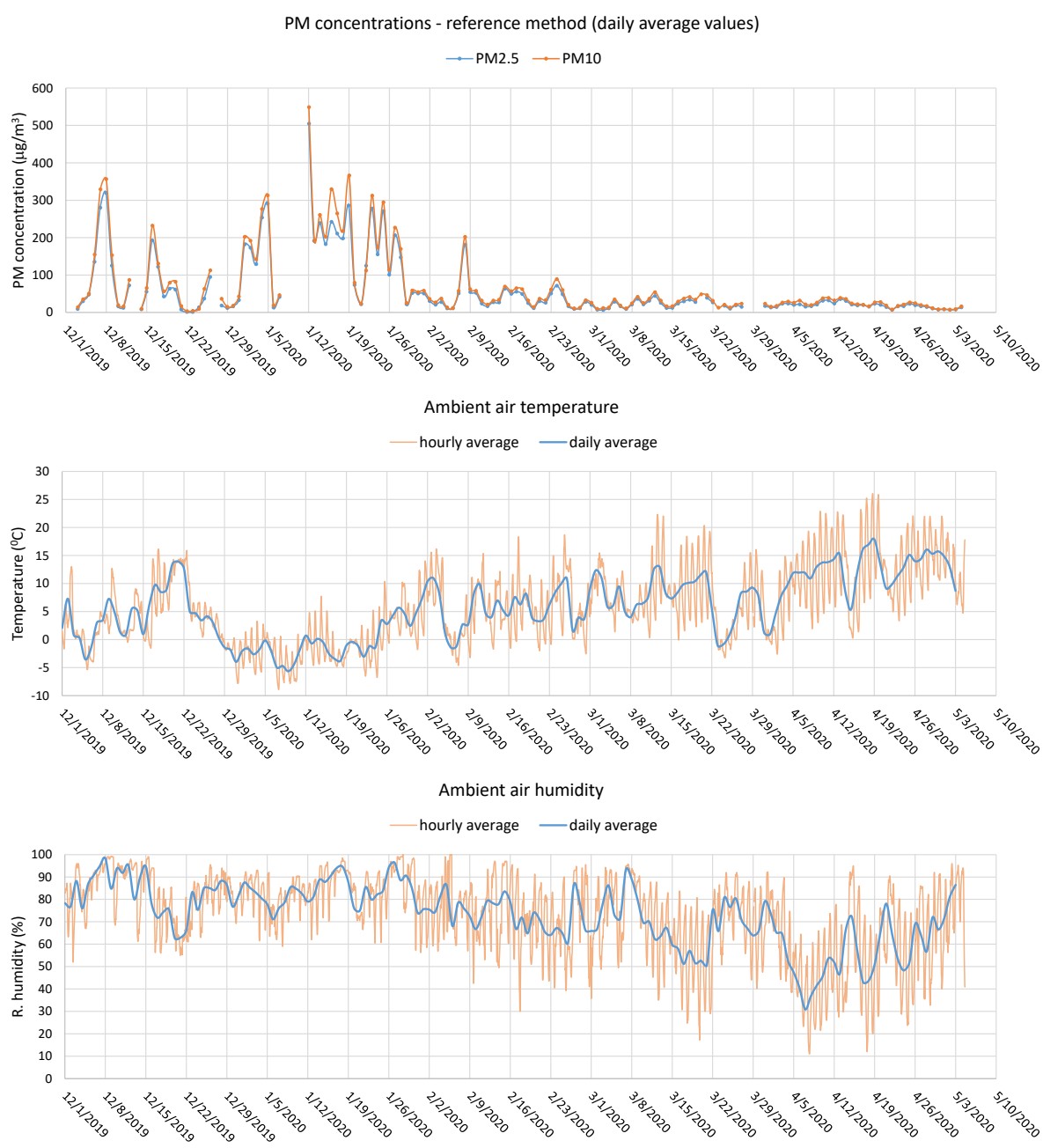

**Figure 4.** Reference PM concentrations with ambient air temperature and humidity.

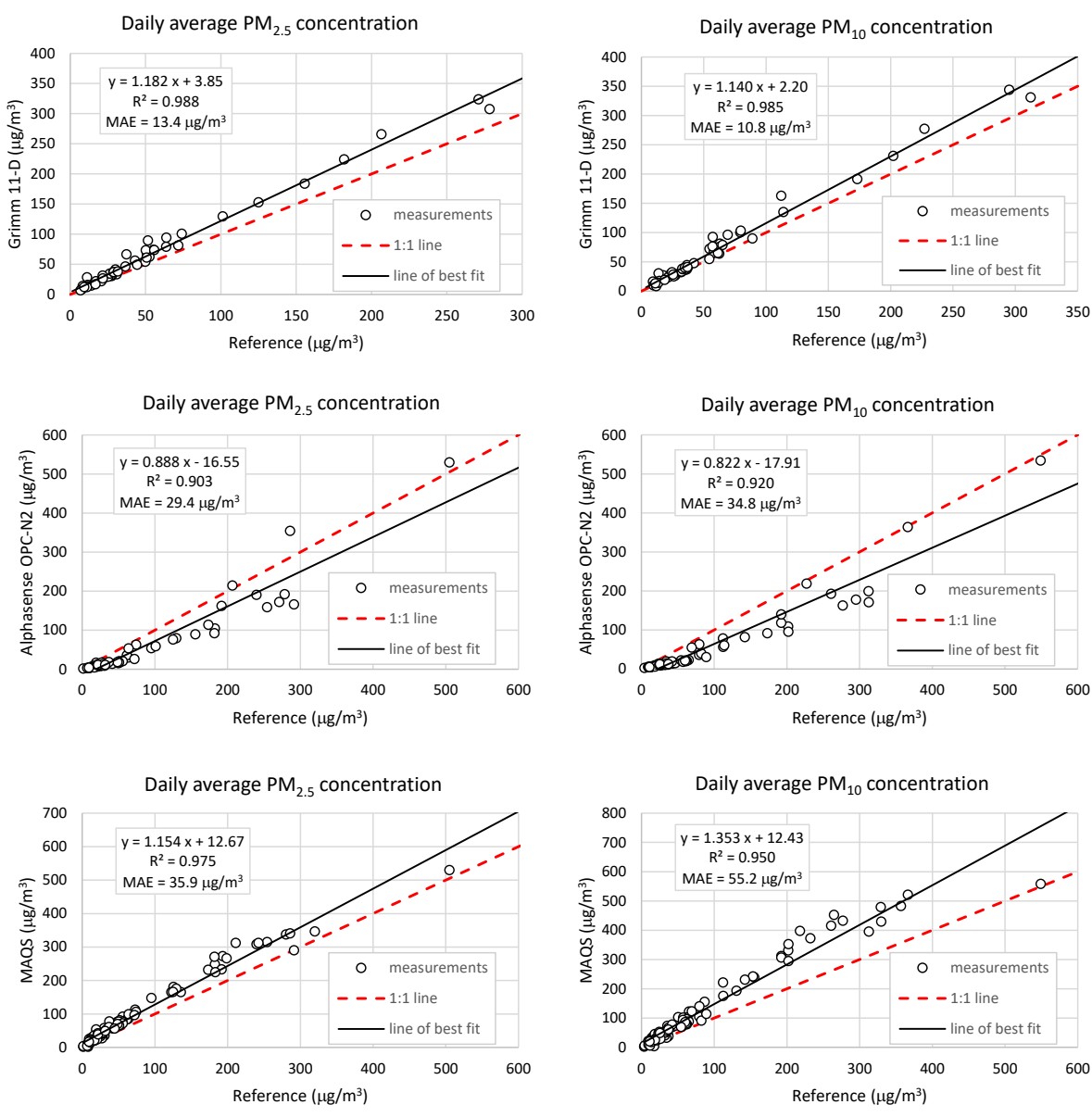

**Figure 5.** OPS performance during the period of strong pollution (12/2/2019–3/12/2020).

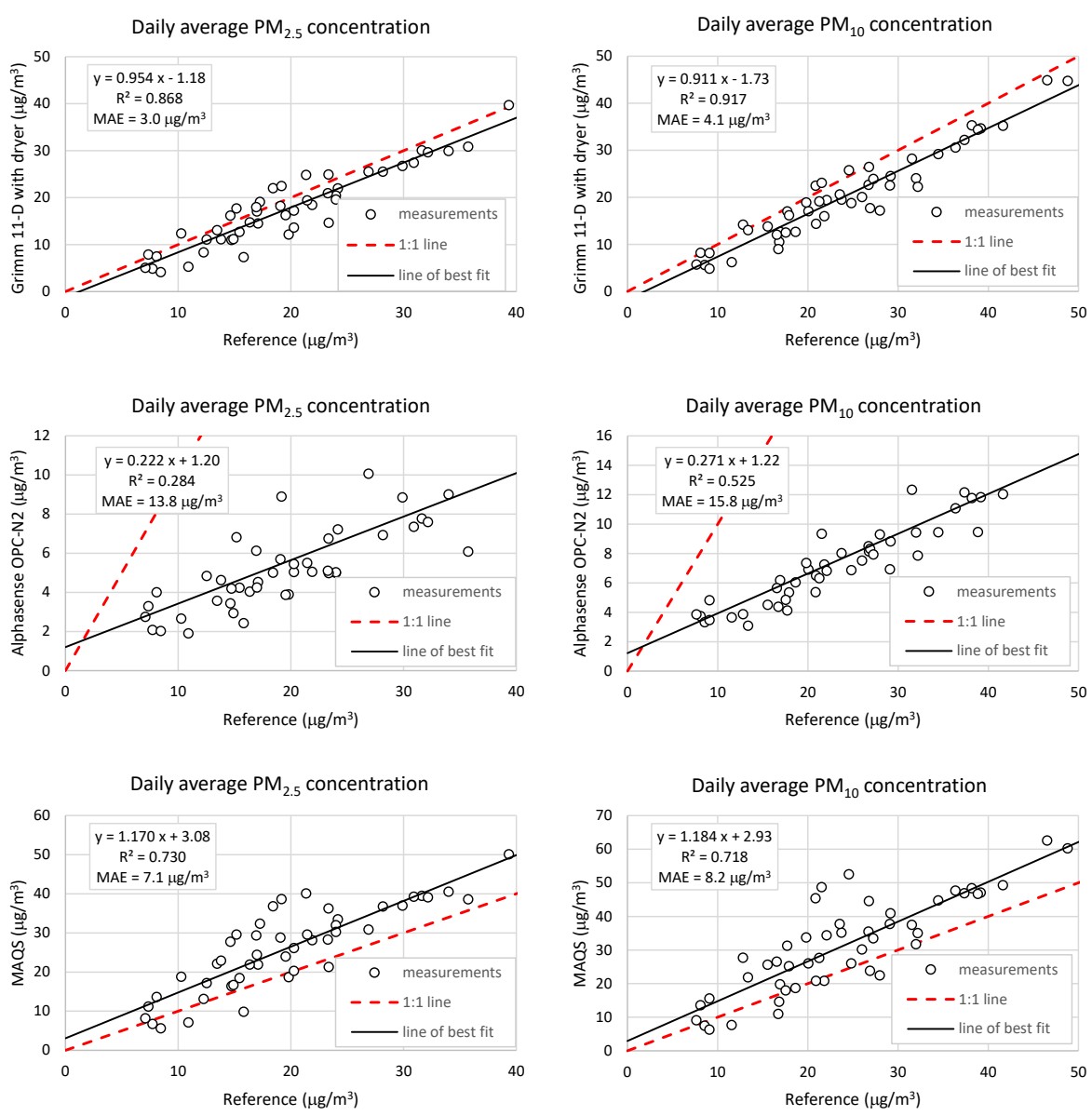

**Figure 6.** OPS performance during the period of mild pollution (3/13/2020–5/4/2020).

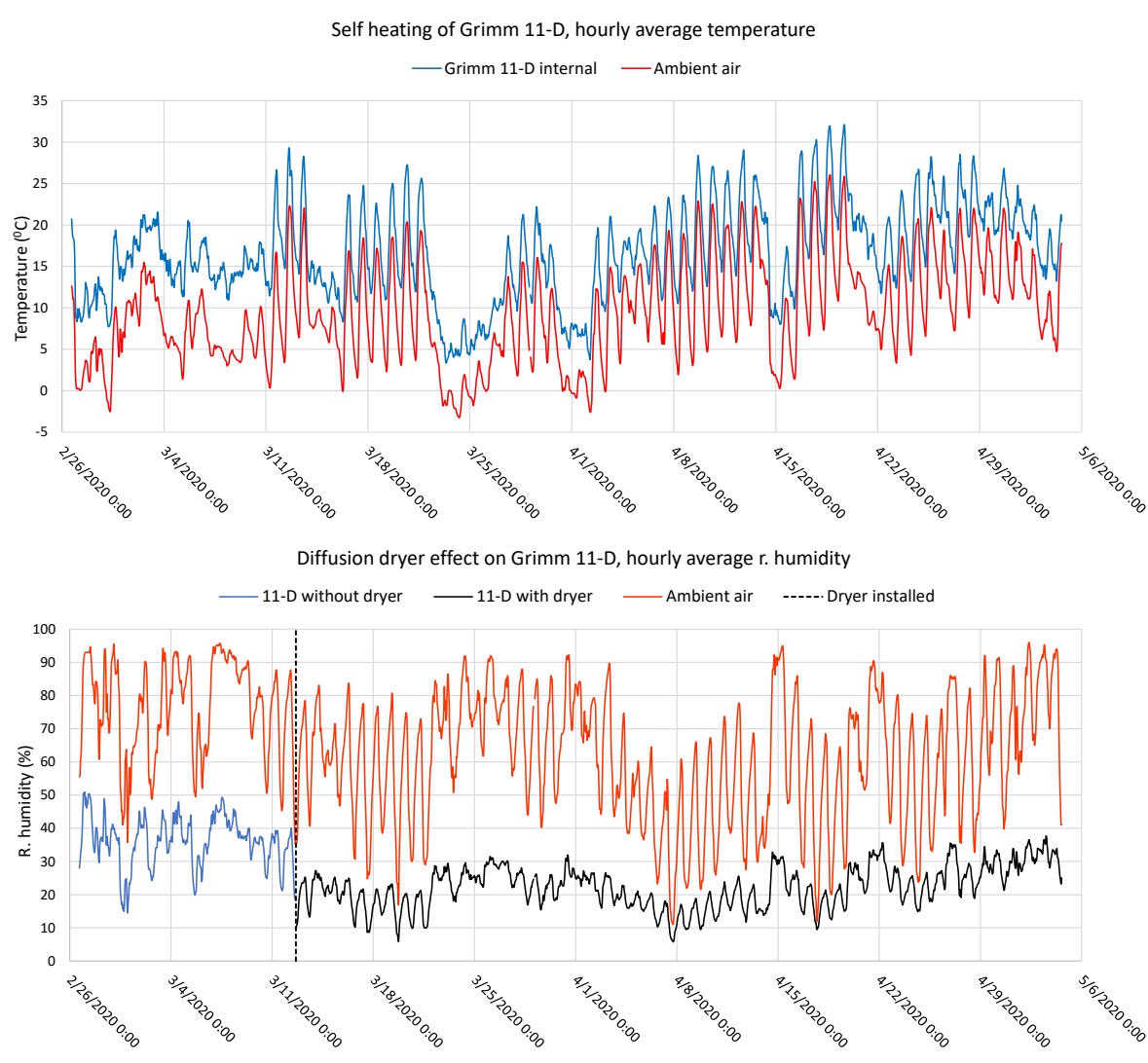

**Figure 7.** Self-heating and diffusion dryer effect of Grimm 11-D. Dryer was installed on 3/12/2020 15:30.

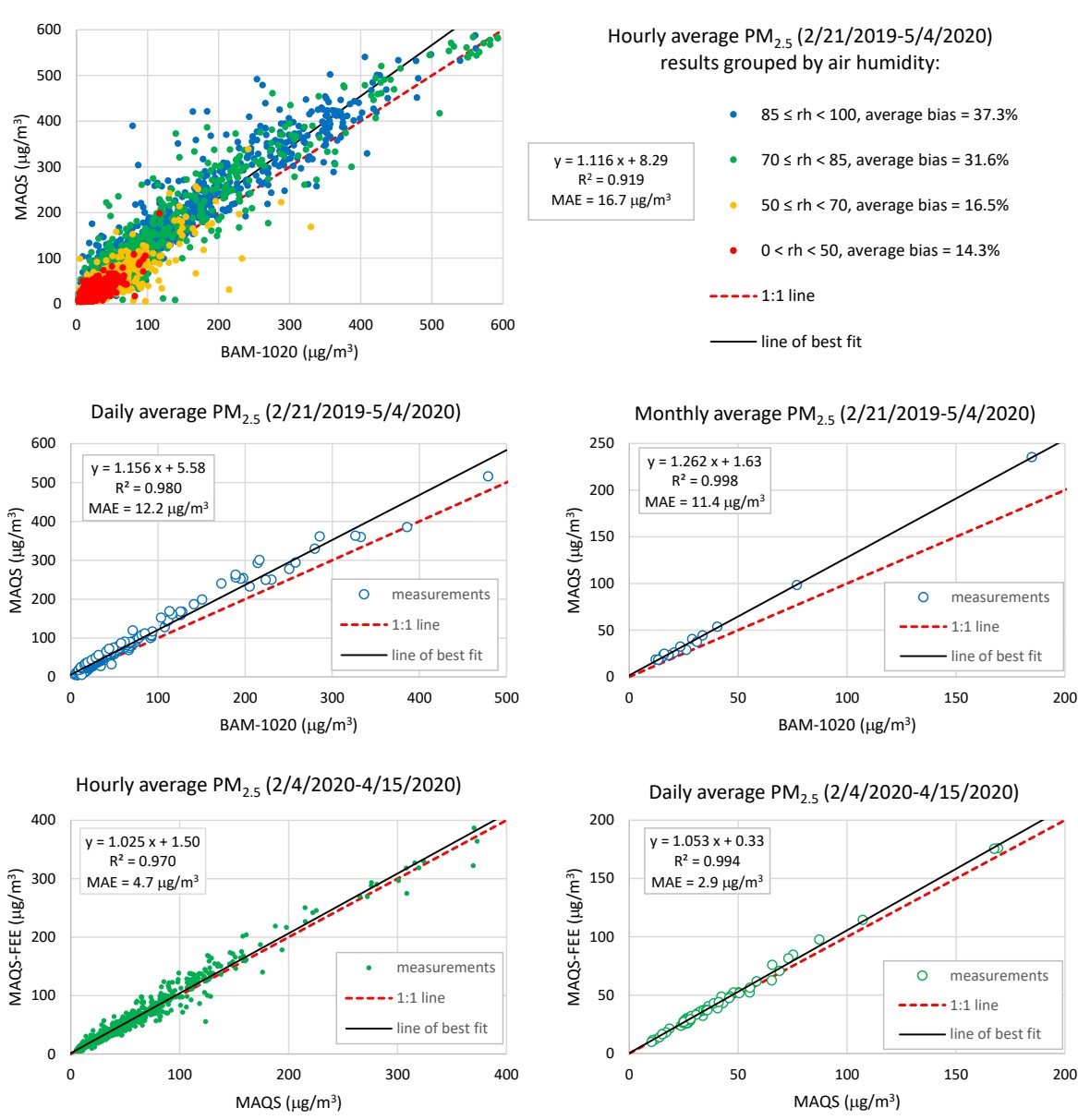

**Figure 8.** Long-term comparisons of MAQS sensor with BAM operated by US EPA at the nearby location: hourly, daily, monthly average values and comparison of hourly and daily average values of two MAQS sensors: first one (MAQS) at our main facility and second one (MAQS-FEE) at Faculty of Electrical Engineering in the immediate vicinity of BAM-1020.

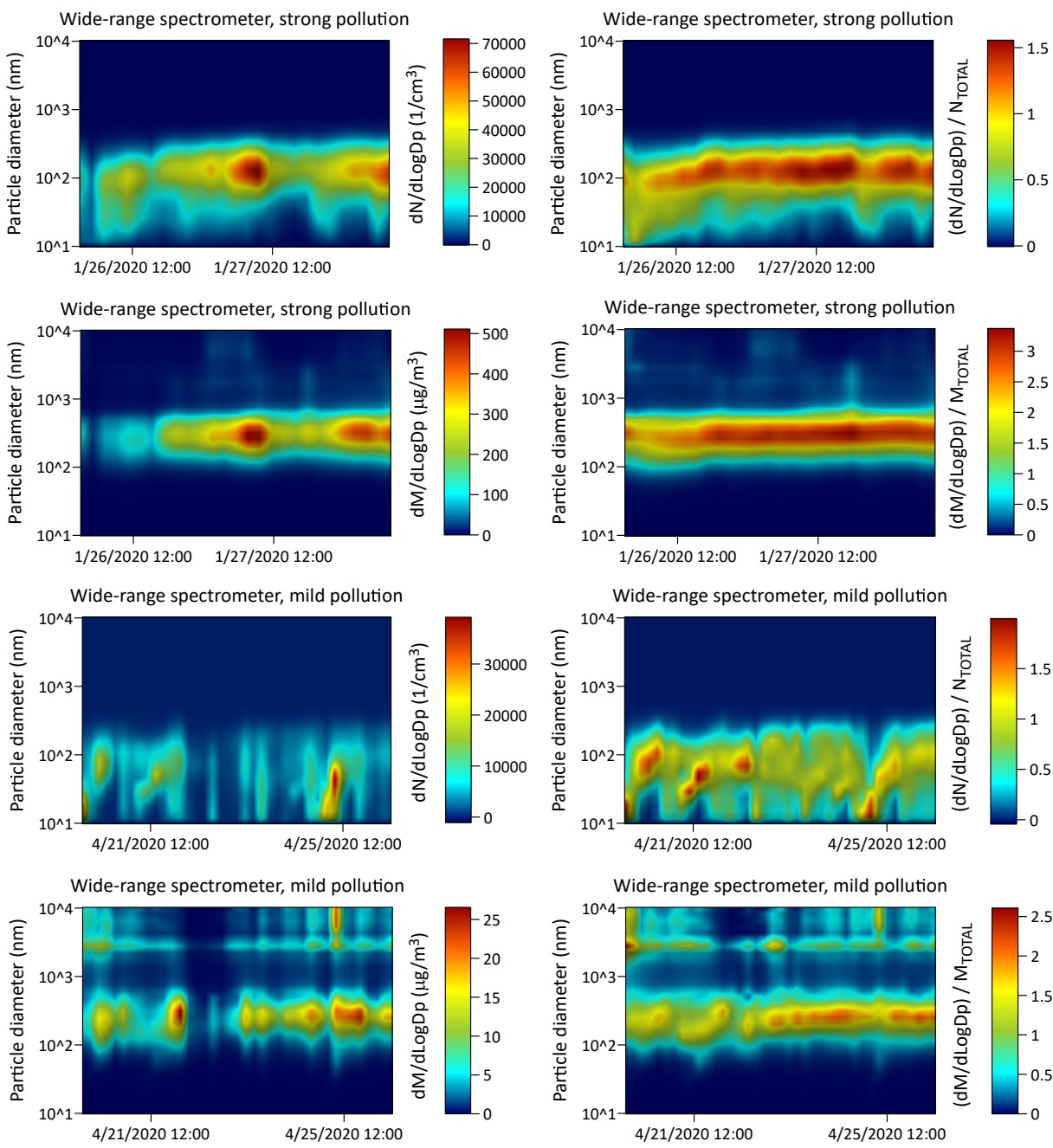

**Figure 9.** Wide-range spectrometer (SMPS+11D), hourly average values. Relative density 1.65 was applied on SMPS to calculate mass of particles.

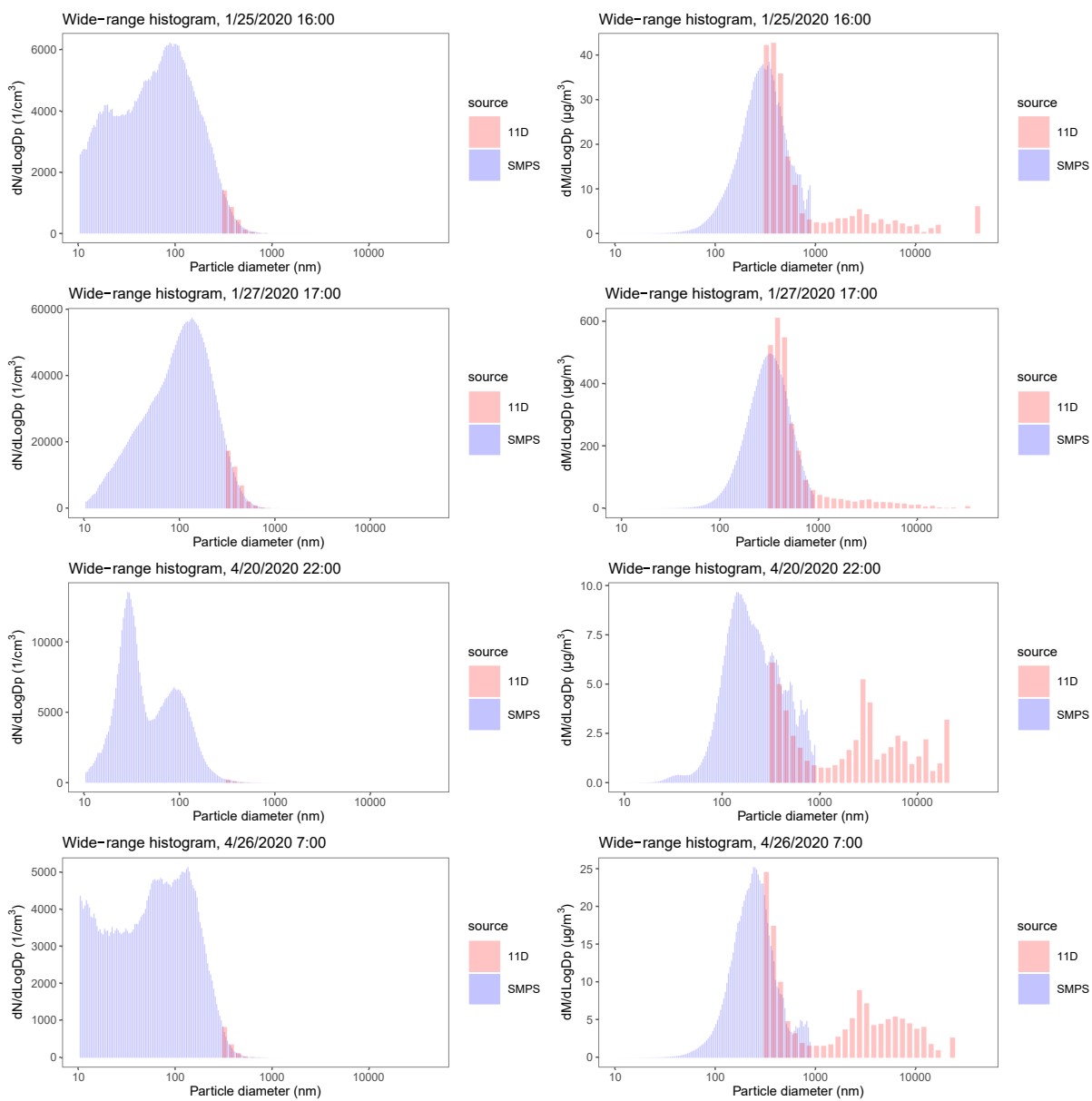

**Figure 10.** Wide-range histograms, hourly average values. Relative density 1.65 was applied on SMPS to calculate mass of particles.

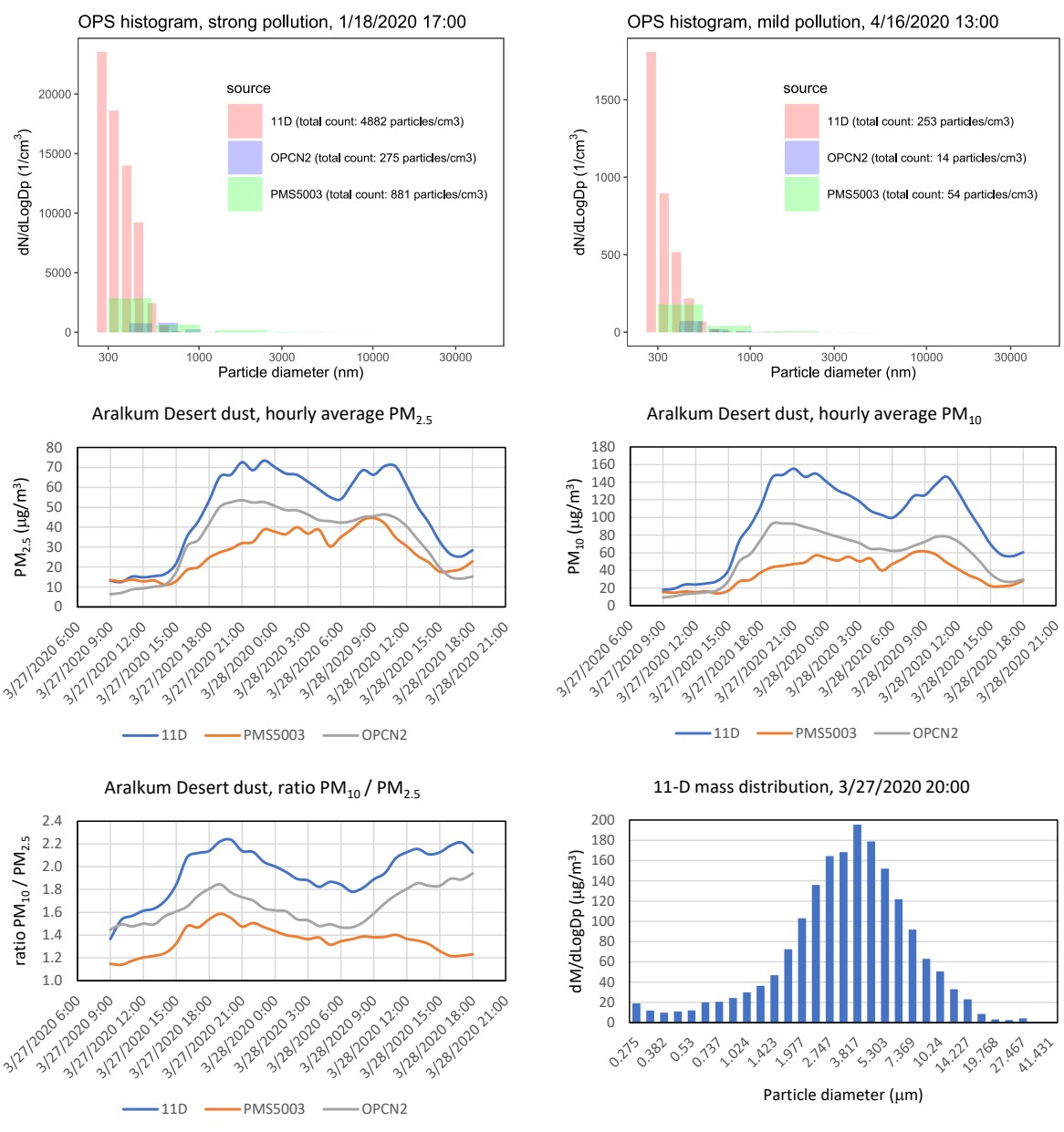

**Figure 11.** OPS histograms and Aralkum Desert dust episode, hourly average values.