# Peer review of "Evaluation of optical particulate matter sensors under realistic conditions of strong and mild urban pollution"

_Atmospheric Measurement Techniques, 2020_

## Referee Comment (RC1) · Anonymous Referee #2 · 4 Aug 2020

General comments: This paper focuses on the performance of 3 OPS devices in a highly polluted area. I think this paper will be helpful to the sensor/air monitoring community as it is at higher ambient concentrations than much previous work with a suite of collocated reference measurements. The authors present a highly valuable dataset. Overall, this is a nice paper with scientific significance, good presentation quality and a few changes to statistical methods/discussion and discussion of previous work will strengthen the scientific quality.

Specific comments:

Line 114: Can you provide any justification as to why you used the AE

channel? The two channels have a nonlinear relationship (Tryner 2020 https://doi.org/10.1016/j.atmosenv.2019.117067) and I wonder if this is some of the reason you have underestimation at high concentration (Figure 3).

The authors have a heavy reliance on R2 throughout this paper even though it is well known that this is not the best comparison between measurement methods (Bland & Altman "STATISTICAL METHODS FOR ASSESSING AGREEMENT BETWEEN TWO METHODS OF CLINICAL MEASUREMENT" Lancet, 1986; i: 307-310). They do also discuss bias (% difference) but I think it would also be helpful to not rely so heavily on discussion of R2 and add another metric of scatter MAE (or RMSE or another metric the authors prefer).

Line 159-161: I don't think this paragraph provides enough details to understand how you calculated this. I'm guessing this is 3 standard deviations but of what? Just zero concentration experienced in the field? Please elaborate as I think these results will be particularly of interest to the field. It seems like the Bulot paper reports LOD from a bunch of previous work with LCS not just PMS5003/N-2 it might be helpful to strengthen the discussion here. More recent work has also explored the LLOD of PMS5003 sensors (e.g. Magi 2019 https://doi.org/10.1080/02786826.2019.1619915). Also did you want to provide any details on what you do with data below the LLOD (throw out, replace, etc)?

Line 177: Also see recent paper on PMS5003 and large particles that may be helpful (Kosmopoulos 2020 https://doi.org/10.1016/j.scitotenv.2020.141396)

The discussion of previous work appears fairly limited. It would be helpful to discuss how the high bias of the PMS5003 and low bias of the OPC-N2 and overall performance compare to studies in other locations as both these devices have been studied fairly extensively.

Section 3.3: You only discuss the Humidity influence on the Grimm it would be helpful to discuss the influences on the PMS5003 and OPC-N2 as well

It seems like you also have the opportunity to discuss the influence of particle size distribution on the performance of the OPC-N2 and PMS5003 but you have limited your discussion to the Grimm. You mention this briefly in lines 245-249 but it seems like instead of just commenting that small particles could be an issue you can look to see of the OPC-N2 is specifically underestimating more because more of the particles are too small. In addition, both the OPC-N2 and PMS5003 have binned data that could be discussed.

Technical corrections:

Line 30: grammatical error "equipped with BAMs" and you should probably spell out what BAM stands for the first time you use it.

Line 123,221: missing m on Grimm

Line 181: It may be helpful to mention the figure earlier on in the paragraph before discussing the results so that readers can look at the figure and follow along.

---

## Referee Comment (RC2) · Anonymous Referee #3 · 8 Sep 2020

The manuscript describes the evaluation of some optical particulate matter sensors in high and low pollution episodes. The topic of very limited novelty as some of the sensors have already been extensively tested in the peer reviewed literature. 2 of the sensors are already not commercially available anymore and follow-up models are being sold. Novelty and generalizable findings would need to be emphasized because right now, there appears little true scientific discussion on fundamentals that would easily be transposable to justify publication of the manuscript on essentially outdated sensors.

The authors should address the following issues

[Figure]

- What is the true novelty here and insights that were not already documented in the existing papers on the Grimm, the alphasense or the PMS?

- You use in the comparison figures linear regressions with non-zero intercepts, some of these intercepts are substantial! >10 ug/m3 for PM10 (figure 3) both positive and negative. This needs to be explained.

- Overall for all the comparison figures, why not indicate a 1:1 line and please do a deeper analysis. It looks like these figures mostly show non linearity with at low concentrations most data points above the lien and at high below or vice versa. There seems to be clear non linearity without any discussion, instead these weird linear regressions with intercepts that are not explained. Even weirder that the authors acknowledge in the text that there is non linearity likely.

- The Alphasense is now on version OPC-N3 and it is hard to find information on earlier version idem on the PMS5003, they are now at PMS7003. Could you comment if you expect the observations here to be transposable otherwise they are useless.

- The description of the PMS device seems very speculative?  This is very weird when a simple google gives clear descriptions of the device (https://www.aqmd.gov/docs/default-source/aq-spec/resources-page/plantower-pms5003-manual_v2-3.pdf )

- The discussion of RH impacts is very cursory and given how big the issue is, it would be important to see how results of the sensors agree or disagree as a function of RH. Here it would be critical to discuss that the gravimetric measurements are done at a given RH but how does this RH compare to the sensor measurements.

- Can you comment that you are running the sensors close to their technical specs (95% RH) also at times you actually do run the Grimm D11 outside of specs as the Grimm specs say temperatures above 4 degC (although you also seem to heat the inlet) this is not very clear.

- The introduction needs serious revision. Particulate matter and aerosol is not the same thing. Please eliminate all discussion of aerosol as aerosol is the particles and the gases

- The introduction is very narrowly focused and does not discuss things like the use of TEOM in networks. Also some of the statements should clearly be supported by references

- You are very non quantitative and non rigorous in the text and very imprecise., This needs substantial improvement. E.g. L81: what is meant by extremely high? L85 you know exactly what your lower size limit is, so please state it, L 166 what is mean by a "good" correlation?

- The statistical discussion totally lack rigor. L 262 "the correlations for hourly, daily and monthly average values of PM2.5 are 0.919, 0.980 and 0.998, respectively" what does this mean? Followed by "with absolute values overestimated by 20% on average " how was this obtained? Where is the data? This is not obvious from Fig 8 at all?

- the abstract should not read like an experiential section with study dates etc. These details should not go there instead it should contain quantitative results form the paper.

———————————————

---

## Author Comment (AC2) · 13 Oct 2020

**Response to Anonymous Referee #3**

We would like to thank the reviewer for the thoughtful and constructive examination of our paper! Please find below our responses to each comment individually and please note that:

- **Blue bold font** represents comments of the Referee,

5 - Black regular font represents the response to each referee's comment,

-  represents removed text from the manuscript according to referee's comment,

- Red font represents added text in the manuscript according to referee's comments.

**The manuscript describes the evaluation of some optical particulate matter sensors in high and low pollution episodes. The topic of very limited novelty as some of the sensors have already been extensively tested in the peer reviewed**
10 **literature. 2 of the sensors are already not commercially available anymore and follow-up models are being sold. Novelty and generalizable findings would need to be emphasized because right now, there appears little true scientific discussion on fundamentals that would easily be transposable to justify publication of the manuscript on essentially outdated sensors.**

Response: We couldn't agree with this general comment. These sensors are not outdated and they are commercially available:
15 as of today, both Grimm 11-D and Plantower PMS5003 are fully available, while Alphasense OPC-N2 is available in limited quantities for existing customers.

**The authors should address the following issues**
**- What is the true novelty here and insights that were not already documented in the existing papers on the Grimm, the alphasense or the PMS?**
20 Response: the true novelty here is the range of ambient air concentrations, which is much wider than any previous work that we could find in existing articles. This unique dataset is relevant for many countries in Eastern Europe, which share the same problem: air pollution with PM as a dominant component. Performance of optical PM sensors greatly depends on composition of PM. In Eastern Europe many houseolds still use coal as the source of energy for heating. Furthermore, network of govermnetal air quality measuremt stations is relativelae sparse. Considering these facts, it is very importnat to systematicaly
25 analyze performance of OPS in this region. Furthermore, wide-range spectrometer, which consists of SMPS and OPC 11-D is for the first time successfully applied in realistic scenario under such ambient air conditions.

**- You use in the comparison figures linear regressions with non-zero intercepts, some of these intercepts are substantial! >10 ug/m3 for PM10 (figure 3) both positive and negative. This needs to be explained.**
Response: yes, indeed, the intercept values for the lines of best fit in Fig. 3. in the manuscript vary from -17.9 $\mu$g/m$^3$ to
30 12.7 $\mu$g/m$^3$. But if we take into account that the range of measurements is from 0 to 600 $\mu$g/m$^3$, these intercepts are relatively small. Please have a look at revised pictures where we have added 1:1 line, according to your suggestion. From these graphs we can see that these intercepts are visually close to the ideal situation (1:1 line).

**Overall for all the comparison figures, why not indicate a 1:1 line and please do a deeper analysis. It looks like these figures mostly show non linearity with at low concentrations most data points above the lien and at high below or**
35 **vice versa. There seems to be clear non linearity without any discussion, instead these weird linear regressions with intercepts that are not explained. Even weirder that the authors acknowledge in the text that there is non linearity likely.**
Response: thank you very much for this suggestion! We have new graphs below with included 1:1 line. Regarding the linearity, actually it is surprisingly good across the wide range of measurements for all 3 tested devices. Figures 3 and 8 show
40 some non-linear tendency of tested sensors only above 300 $\mu$g/m$^3$ of PM$_{2.5}$.

[Figure]

**Figure 5.** OPS performance during the period of strong pollution (12/2/2019–3/12/2020).

[Figure]

**Figure 6.** OPS performance during the period of mild pollution (3/13/2020–5/4/2020).

[Figure]

**Figure 8.** Long-term comparisons of MAQS sensor with BAM operated by US EPA at the nearby location: hourly, daily, monthly average values and comparison of hourly and daily average values of two MAQS sensors: first one (MAQS) at our main facility and second one (MAQS-FEE) at Faculty of Electrical Engineering in the immediate vicinity of BAM-1020.

**- The Alphasense is now on version OPC-N3 and it is hard to find information on earlier version idem on the PMS5003, they are now at PMS7003. Could you comment if you expect the observations here to be transposable otherwise they are useless.**

Response: OPC-N3 is a successor to OPC-N2. By analyzing the specifications of both, we expect that these results are applicable to OPC-N3 as well. Major differences between OPC-N3 and OPC-N2 are internal temperature and humidity sensor in OPC-N3 and slightly lower detection limit, 350 nm (N3) instedad of 380 nm (N2). While these are useful features, they certainly don't make OPC-N2 obsolete.

Plantower PMS7003 is a miniaturized version of PMS5003. The only advantage of PMS7003 is its smaller size. Because of that PMS7003 is preferred if the size is critical, for example the AirBeam project (Mukherjee, 2017, Sensors). But for our research PMS5003 is more appropriate, because it is more rugged and it has larger intake fan. PurpleAir (Tryner, 2020, AE) is good example of integration of PMS5003 into the network of sensors. We have many laboratory test results of PMS7003 and PMS5003 and conclusion is clear: these sensors give the same results. Please have a look at the results of one of our tests, where we compare these two sensors. Burning chamber with incense scents as the source of PM was used.

[Figure]

**Comparison of PMS7003 and PMS5003**

**- The description of the PMS device seems very speculative? This is very weird when a simple google gives clear descriptions of the device ( https://www.aqmd.gov/docs/default-source/aq-spec/resources-page/plantowerpms5003-manual_v2-3.pdf )**

Response: we have that file, which contains specifications of PMS5003. However, some important information about the sensor is not published there. For example, there are two modes (SM and AE) and the Manufacturer did not publish what is the difference. But we agree that our description of PMS5003 is not good, and we have changed it.

 It uses red semiconductor laser, photodetector at $90^0$ scattering angle (Kuula, 2020) and 32-bit processor (Cypress CY8C4245, 48 MHz). According to (Tanzer, 2019) PMS5003 is a nephelometer, not the particle counter.

**- The discussion of RH impacts is very cursory and given how big the issue is, it would be important to see how results of the sensors agree or disagree as a function of RH. Here it would be critical to discuss that the gravimetric measurements are done at a given RH but how does this RH compare to the sensor measurements.**

Response: according to requirements of the standard EN 12341:2014, all filters were conditioned in our gravimetric laboratory at relative humidity between 45% and 50%, and temperature between 19 and 21 $^0$C. Without dryers on all devices, we can not control these parameters outside. However, this campaign provided us with enough measurements to draw some important conclusions about humidity influence on measurements from OPS. This text is added in section 2:

According to requirements of the standard, all filters were conditioned at relative humidity between 45% and 50%, and temperature between 19 and 21 $^0$C.

This text is added in section 3.3:

Figure 8 shows the long-term (13.5 months) comparison of MAQS and BAM-1020 with time resolution of 1 hour, together with measured values of ambient air humidity. By averaging all this data we can estimate the influence of humidity on the MAQS sensor: if we sort the measurements by humidity, subset of points where humidity is below 50% has average bias of

14.3%, for humidity range 50%-70%, bias is 16.5%; for humidity range 70%-85% bias is 31.6% and for humidity rang 85%-100% bias is 37.3%. If we subtract bias of least humidity subset from bias of highest humidity subset, we can estimate that humidity influence adds up to 23% on PM2.5 readings from MAQS sensor, which is similar result to the analysis of humidity influence on our 11-D with dryer installed. While this influence can not be neglected, it is still relatively modest. Reason for this is the composition of particles, where we have mostly fine particles below 300 nm, for which hygroscopic growth is less pronounced (Kosmopoulos 2020).

**Can you comment that you are running the sensors close to their technical specs (95% RH) also at times you actually do run the Grimm D11 outside of specs as the Grimm specs say temperatures above 4 degC (although you also seem to heat the inlet) this is not very clear.**

Response: primary aim of this research is the evaluation of instruments in realistic scenario. That includes wide range of all operating parameters. During the campaign we followed all recommendations of manufacturers of devices, especially Grimm, to ensure that instruments are running normally. Rigorous quality assurance procedure was used. All measurements below LLOD, outside of the specifications and with error or warning codes in the logs were discarded.

**The introduction needs serious revision. Particulate matter and aerosol is not the same thing. Please eliminate all discussion of aerosol as aerosol is the particles and the gases**

Response: we accept this. Instead of "aerosol", "particulate matter" or other appropriate term will be used consistently through the entire manuscript.

**The introduction is very narrowly focused and does not discuss things like the use of TEOM in networks. Also some of the statements should clearly be supported by references**

Response: we agree that network of TEOMs is an interesting topic, but it is not related to this work. TEOM is not an optical scattering device.

**You are very non quantitative and non rigorous in the text and very imprecise., This needs substantial improvement. E.g. L81: what is meant by extremely high? L85 you know exactly what your lower size limit is, so please state it, L 166 what is mean by a "good" correlation?**

Response: L81 "extremely high" refers to measured value of PM2.5 concentration up to 504.9 $\mu$g/m$^3$. That is extremely high concentration of $PM_{2.5}$. For example, in US AQI categorization, values of $PM_{2.5}$ over 500 $\mu$g/m$^3$ are beyond air quality index scale. We accept objection about lower size limit in L88, so instead

we have

It can detect particles with diameters from 10 nm up to 1 $\mu$m.

$R^2$ coefficients from 0.90 to 0.99 represent very good correlation in this context (comparison of optical devices to the reference gravimetric method).

**The statistical discussion totally lack rigor. L 262 "the correlations for hourly, daily and monthly average values of PM2.5 are 0.919, 0.980 and 0.998, respectively" what does this mean? Followed by "with absolute values overestimated by 20% on average " how was this obtained? Where is the data? This is not obvious from Fig 8 at all?**

Response: Fig 8 is redrawn completely, and now you can see these average bias values on the figure as a function of air humidity. The underlying datasets for this publication are available at

https://doi.org/10.5281/zenodo.3897379

Here are the text changes:

Based on 13.5 months of continuous comparison of MAQS and BAM-1020, hourly average values give $R^2$ coefficient 0.919 and MAE 16.7 $\mu$g/m$^3$. Daily average values produce $R^2$ coefficient 0.980 and MAE 12.2 $\mu$g/m$^3$, while the monthly average values give $R^2 = 0.998$ and MAE $= 11.4$ $\mu$g/m$^3$ (Figure 8).

**- the abstract should not read like an experiential section with study dates etc. These details should not go there instead it should contain quantitative results form the paper.**

Response: we accept that, here is the new abstract:

135     In this paper we evaluate characteristics of three optical particulate matter sensors/sizers (OPS): high-end spectrometer 11-D (Grimm, Germany), low-cost sensor OPC-N2 (Alphasense, United Kingdom) and in-house developed MAQS (Mobile Air Quality System) which is based on another low-cost sensor – PMS5003 (Plantower, China), under realistic conditions of strong and mild urban pollution. Results were compared against a reference gravimetric system, based on Gemini (Dadolab, Italy), $2.3\,\mathrm{m^3/h}$ air sampler, with two channels (simultaneously measuring $PM_{2.5}$ and $PM_{10}$ concentrations). The measurements were

140     performed in Sarajevo, the capital of Bosnia-Herzegovina, from December 2019 until May 2020.  This interval is divided into period 1 - strong pollution and period 2 - mild pollution. The city of Sarajevo is one of the most polluted cities in Europe in terms of aerosols: the average concentration of $PM_{2.5}$ during the period 1 was 83 $\mu$g/m$^3$, with daily average values exceeding 500 $\mu$g/m$^3$. During period 2, the average concentration of $PM_{2.5}$ was 20 $\mu$g/m$^3$. These conditions represent

145     a good opportunity to test optical devices against reference instrument in a wide range of ambient particulate matter (PM) concentrations. The effect of an in-house developed diffusion dryer for 11-D is discussed as well. In order to analyze the mass distribution of  particles, a scanning mobility particle sizer (SMPS), which together with the 11-D spectrometer gives the full spectrum from nanoparticles of diameter 10 nm to coarse particles of diameter 35 $\mu$m, was used. All tested devices showed excellent correlation with the reference instrument in period 1, with $R^2$ values between 0.90 and 0.99 for

150     daily average PM concentrations. However, in period 2, where the range of concentrations was much narrower, $R^2$ values decreased significantly, to values from 0.28 to 0.92. We have also included results of a 13.5 month long-term comparison of our MAQS sensor with a nearby beta attenuation monitor (BAM) 1020 (Met One Instruments, USA) operated by the United States Environmental Protection Agency (US EPA), which showed similar correlation and no observable change of performance over time.

---

## Author Response (AR1)

**Response to Anonymous Referee #2**

We would like to thank the reviewer for the thoughtful and constructive examination of our paper! Please find below our responses to each comment individually and please note that:

- **Blue bold font** represents comments of the Referee,

- Black regular font represents the response to each referee's comment,

-  represents removed text from the manuscript according to referee's comment,

- Red font represents added text in the manuscript according to referee's comments.

**General comments: This paper focuses on the performance of 3 OPS devices in a highly polluted area. I think this paper will be helpful to the sensor/air monitoring community as it is at higher ambient concentrations than much previous**
10 **work with a suite of collocated reference measurements. The authors present a highly valuable dataset. Overall, this is a nice paper with scientific significance, good presentation quality and a few changes to statistical methods/discussion and discussion of previous work will strengthen the scientific quality.**
Response: Thank you for the general comments! We will try to implement your suggestions.

**Line 114: Can you provide any justification as to why you used the AE channel? The two channels have a nonlinear**
15 **relationship (Tryner 2020 https://doi.org/10.1016/j.atmosenv.2019.117067) and I wonder if this is some of the reason you have underestimation at high concentration (Figure 3).**
Response: The manufacturer of PMS5003 sensor, Plantower, has not explained publicly what is the difference between AE and SM modes, but in private communication they explicitly confirmed that AE is the correct channel for ambient air measurements, while the SM mode is recommended for industrial production workplaces (metal particles or other higher
20 density particles). We have performed laboratory test of these two modes on PMS5003 using the burning chamber and incense scents as the source of PM, and here are the results:

[Figure]

**Figure 2.** AE and SM modes of PMS5003 sensor.

The relationship between SM and AE modes can be deduced from our test results:

$$
SM_{PM2.5} = \begin{cases} AE_{PM2.5} \text{ for } AE_{PM2.5} \leq 30 \\ \text{nonlinear for } 30 < AE_{PM2.5} \leq 50 \\ 1.5 \times AE_{PM2.5} \text{ for } AE_{PM2.5} > 50 \end{cases}
$$

$$
SM_{PM10} = \begin{cases} AE_{PM10} \text{ for } AE_{PM10} \leq 43 \\ \text{nonlinear for } 43 < AE_{PM10} \leq 77 \\ 1.5 \times AE_{PM10} \text{ for } AE_{PM10} > 77 \end{cases}
$$

Investigation of Tryner et al (2020) https://doi.org/10.1016/j.atmosenv.2019.117067 has limited range of PM concentrations, where we can't see this third segment of direct proportionality between SM and AE mode for higher PM concentrations. In conclusion, for low concentrations these two modes are the same, in the narrow, intermediate range the relationship between SM and AE mode is nonlinear, while for higher concentrations SM mode simply gives 50% higher value than AE mode, for both PM2.5 and PM10. If applied to our measurements, SM mode would overestimate actual PM concentrations more than AE mode. These results are added in section 2.

**The authors have a heavy reliance on R2 throughout this paper even though it is well known that this is not the best comparison between measurement methods (Bland & Altman "STATISTICAL METHODS FOR ASSESSING AGREEMENT BETWEEN TWO METHODS OF CLINICAL MEASUREMENT" Lancet, 1986; i: 307-310). They do also discuss bias (% difference) but I think it would also be helpful to not rely so heavily on discussion of R2 and add another metric of scatter MAE (or RMSE or another metric the authors prefer).**

Response: we agree completely. For all graphs on figures 3, 4 and 8 in manuscript we have added MAE value and 1:1 line. These MAE values are discussed in sections 3.1, 3.2 and 3.6. Furthermore, Figure 8 is modified for some later discussion on humidity influence. Here are the new graphs:

[Figure]

**Figure 5.** OPS performance during the period of strong pollution (12/2/2019–3/12/2020).

[Figure]

**Figure 6.** OPS performance during the period of mild pollution (3/13/2020–5/4/2020).

[Figure]

**Figure 8.** Long-term comparisons of MAQS sensor with BAM operated by US EPA at the nearby location: hourly, daily, monthly average values and comparison of hourly and daily average values of two MAQS sensors: first one (MAQS) at our main facility and second one (MAQS-FEE) at Faculty of Electrical Engineering in the immediate vicinity of BAM-1020.

**Line 159-161: I don't think this paragraph provides enough details to understand how you calculated this. I'm guessing this is 3 standard deviations but of what? Just zero concentration experienced in the field? Please elaborate as I think these results will be particularly of interest to the field. It seems like the Bulot paper reports LOD from a bunch of previous work with LCS not just PMS5003/N-2 it might be helpful to strengthen the discussion here. More recent work has also explored the LLOD of PMS5003 sensors (e.g. Magi 2019 https://doi.org/10.1080/02786826.2019.1619915). Also did you want to provide any details on what you do with data below the LLOD (throw out, replace, etc)?**

Response: we agree and here are the changes:

Standard deviation ($\sigma$) was calculated for periods with near-zero ambient PM concentration and average value of $3\sigma$ is estimated LLoD. For PMS5003 our final estimation is $5\,\mu g/m^3$. The same value is an estimation of (Magi et al., 2020), calculated by averaging segmented regressions and (Bulot et al., 2019) by combining results from several previous studies. This method applied on OPC-N2 yields LLoD of $2\,\mu g/m^3$ and $1\,\mu g/m^3$ for 11-D. For reference gravimetric system LLoD was calculated using the blank filters, which were treated exactly the same way as real samples (except the sampling of particulate matter), and the calculated value of LLoD is $0.7\,\mu g/m^3$. All measurements below LLoD were discarded during the quality assurance phase.

**Line 177: Also see recent paper on PMS5003 and large particles that may be helpful (Kosmopoulos 2020 https://doi.org/10.1016/j.scitotenv.2020.141396)**

Response: this is a very relevant reference, and we have added it in the revised manuscript. Investigation in Patras, Greece with narrower range of ambient PM concentrations and different aerosol composition with more frequent episodes of Sahara dust. PurpleAir (PMS5003) showed high bias (similar to our results) and bad performance during the Sahara Desert dust episode. We have also registered one intensive desert dust episode on 3/27/2020 from the Aralkum Desert. Here is the new text:

However, different conditions were observed on 3/27/2000 when the dust from Aralkum Desert covered part of Europe, including our test location. During this episode, OPC-N2 performed much better than PMS5003, which wasn't able to determine large fraction of coarse particles correctly (Figure 11). Similar observation about PMS5003 was reported by (Kosmopoulos et al., 2020), when Sahara dust covered Greece.

**The discussion of previous work appears fairly limited. It would be helpful to discuss how the high bias of the PMS5003 and low bias of the OPC-N2 and overall performance compare to studies in other locations as both these devices have been studied fairly extensively.**

Response: the following text is added in section 1:

In (Mukherjee et al., 2017) OPC-N2, PMS7003 and 11-R were compared against BAM-1020 during 12 weeks in the Cuyama Valley, California, USA. Grimm 11-R performed well, while both OPC-N2 and PMS7003 (which is a miniaturized version of PMS5003) produced mediocre performance with heavy low bias. PurpleAir (PMS5003) was tested in (Tryner et al., 2020) using laboratory and field tests. High bias of PMS5003 was observed. In (Magi et al., 2020) PurpleAir (PMS5003) was analyzed for 16 months in Charlotte, North Carolina, USA against BAM-1022, high bias of PMS5003 that increases with humidity was reported. High mean bias of PurpleAir (PMS5003) was reported in (Kosmopoulos et al., 2020) as well.

**Section 3.3: You only discuss the Humidity influence on the Grimm it would be helpful to discuss the influences on the PMS5003 and OPC-N2 as well.**

Response: we have constructed dryers only for 11-D and SMPS (new Figure 1). The design and construction of dryers for small, low-cost sensors, such as OPC-N2 and PMS5003 is in our plans for future work. From the dataset of this study, we can include ambient air humidity as the parameter in Figure 8.

Figure 8 shows the long-term (13.5 months) comparison of MAQS and BAM-1020 with time resolution of 1 hour, together with measured values of ambient air humidity. By averaging all this data we can estimate the influence of humidity on the MAQS sensor: if we sort the measurements by humidity, subset of points where humidity is below 50% has average bias of 14.3%, for humidity range 50%-70%, bias is 16.5%; for humidity range 70%-85% bias is 31.6% and for humidity rang 85%-100% bias is 37.3%. If we subtract bias of least humidity subset from bias of highest humidity subset, we can estimate that humidity influence adds up to 23% on PM2.5 readings from MAQS sensor, which is similar result to the analysis of humidity influence on our 11-D with dryer installed. While this influence can not be neglected, it is still relatively modest. Reason for

95 this is the composition of particles, where we have mostly fine particles below 300 nm, for which hygroscopic growth is less effective (Kosmopoulos 2020).

[Figure]

(a) Air sampler and Stevenson screen.           (b) Devices under test.

[Figure]

(c) SMPS with dryer.

**Figure 1.** Experimental setup: a) co-located air sampler and Stevenson screen, b) devices under test inside of Stevenson screen: 11-D with dryer, OPC-N2 with SPI adapter and (white-orange) enclosure, MAQS (white enclosure with grey front
100 panel), and c) indoors SMPS with dryer.

**It seems like you also have the opportunity to discuss the influence of particle size distribution on the performance of the OPC-N2 and PMS5003 but you have limited your discussion to the Grimm. You mention this briefly in lines 245-249 but it seems like instead of just commenting that small particles could be an issue you can look to see of the OPC-N2 is specifically underestimating more because more of the particles are too small. In addition, both the OPC-N2 and**
105 **PMS5003 have binned data that could be discussed.**

Response: We have data bins for OPC-N2 and PMS5003. However, there are some doubts if the comparison of data bins from PMS5003 and OPC-N2 to 11-D are appropriate, due to the following reasons:

– these devices are constructed differently, 11-D has hydrodynamic focusing (sheath flow), regulated flow rates and high-performance optics, while the low-cost sensors couldn't count individual particles in the same way

110

- PMS5003 is not a particle counter, it is a nephelometer (Tanzer, 2019) doi.org/10.3390/ijerph16142523

- OPC-N2 has detection limit of 380 nm, PMS5003 from 300 nm, while 11-D can count particles from 250 nm

- when calculating PM mass concentration from individual data bins manufacturers use different weighting factors. Only Alphasense provides values of weighting factors for data bins, assumed particle density for each bin and index of refraction (especially important for correct determination of the particle's diameter). These values for 11-D and PMS5003 are not known.

Taking into account these notes, we present new subsection in the revised manuscript:

**3.5 OPS histograms and Aralkum Desert dust**

All tested OPS have data bins, with different number of channels, as described in section 2. Figure 11 shows histograms that compare data bins from 11-D, OPC-N2 and MAQS on 1/18/2020 (strong pollution) and 4/16/2020 (mild pollution). It should be noted that we compare here data bins from devices with different specifications and category. As expected, 11-D has ability to count particles below 300 nm, which appear in greatest numbers. Counting efficiency of OPC-N2 is investigaetd in laboratory conditions using PSL particles in (Sousan et al., 2016a), and the results were good for particles larger than 0.8 $\mu$m while for particles with diameter of 0.5 $\mu$m OPC-N2 the device showed lower detection efficiency (detection limit of OPC-N2 is 0.38 $\mu$m). In our realistic scenario, dominant contribution to the mass comes from particles much smaller than 0.8 $\mu$m (Figures 9 and 10) which is not favorable to OPC-N2.

Contrary to OPC-N2, PMS5003 has problems with coarse particle, as indicated in laboratory test (Kuula et al., 2020). If the fraction of coarse particles is small and steady, PMS5003 performs much better. Ambient conditions in Bosnia-Herzegovina are such most of the time, since the primary source of PM is combustion of coal and biomass. That could explain why PMS5003 performs better than OPC-N2 most of the time. However, different condition were observed on 3/27/2000 when the dust from Aralkum Desert covered part of Europe, including our test location. During this episode, OPC-N2 performed much better than PMS5003, which wasn't able to determine large fraction of coarse particles correctly (Figure 11). Similar observation about PMS5003 was reported by (Kosmopoulos et al., 2020), when Sahara dust covered Greece.

[Figure]

**Figure 11.** OPS histograms and Aralkum Desert dust episode, hourly average values.

**Technical corrections: Line 30: grammatical error "equipped with BAMs" and you should probably spell out what BAM stands for the first time you use it. Line 123,221: missing m on Grimm Line 181: It may be helpful to mention the figure earlier on in the paragraph before discussing the results so that readers can look at the figure and follow along.**

Response: we accept all these suggestions and appropriate corrections have been made for the revised text.

**Response to Anonymous Referee #3**

We would like to thank the reviewer for the thoughtful and constructive examination of our paper! Please find below our responses to each comment individually and please note that:

145     – **Blue bold font** represents comments of the Referee,

    – Black regular font represents the response to each referee's comment,

    –  represents removed text from the manuscript according to referee's comment,

    – Red font represents added text in the manuscript according to referee's comments.

**The manuscript describes the evaluation of some optical particulate matter sensors in high and low pollution episodes.**
150 **The topic of very limited novelty as some of the sensors have already been extensively tested in the peer reviewed literature. 2 of the sensors are already not commercially available anymore and follow-up models are being sold. Novelty and generalizable findings would need to be emphasized because right now, there appears little true scientific discussion on fundamentals that would easily be transposable to justify publication of the manuscript on essentially outdated sensors.**

155 Response: We couldn't agree with this general comment. These sensors are not outdated and they are commercially available: as of today, both Grimm 11-D and Plantower PMS5003 are fully available, while Alphasense OPC-N2 is available in limited quantities for existing customers.

**The authors should address the following issues**
**- What is the true novelty here and insights that were not already documented in the existing papers on the Grimm,**
160 **the alphasense or the PMS?**

Response: the true novelty here is the range of ambient air concentrations, which is much wider than any previous work that we could find in existing articles. This unique dataset is relevant for many countries in Eastern Europe, which share the same problem: air pollution with PM as a dominant component. Performance of optical PM sensors greatly depends on composition of PM. In Eastern Europe many houseolds still use coal as the source of energy for heating. Furthermore, network
165 of govermnetal air quality measuremt stations is relativelae sparse. Considering these facts, it is very importnat to systematicaly analyze performance of OPS in this region. Furthermore, wide-range spectrometer, which consists of SMPS and OPC 11-D is for the first time successfully applied in realistic scenario under such ambient air conditions.

**- You use in the comparison figures linear regressions with non-zero intercepts, some of these intercepts are substantial!**
**>10 ug/m3 for PM10 (figure 3) both positive and negative. This needs to be explained.**
170 Response: yes, indeed, the intercept values for the lines of best fit in Fig. 3. in the manuscript vary from -17.9 $\mu$g/m$^3$ to 12.7 $\mu$g/m$^3$. But if we take into account that the range of measurements is from 0 to 600 $\mu$g/m$^3$, these intercepts are relatively small. Please have a look at revised pictures where we have added 1:1 line, according to your suggestion. From these graphs we can see that these intercepts are visually close to the ideal situation (1:1 line).

**Overall for all the comparison figures, why not indicate a 1:1 line and please do a deeper analysis. It looks like these**
175 **figures mostly show non linearity with at low concentrations most data points above the lien and at high below or vice versa. There seems to be clear non linearity without any discussion, instead these weird linear regressions with intercepts that are not explained. Even weirder that the authors acknowledge in the text that there is non linearity likely.**

Response: thank you very much for this suggestion! We have new graphs below with included 1:1 line. Regarding the
180 linearity, actually it is surprisingly good across the wide range of measurements for all 3 tested devices. Figures 3 and 8 show some non-linear tendency of tested sensors only above 300 $\mu$g/m$^3$ of PM$_{2.5}$.

[Figure]

**Figure 5.** OPS performance during the period of strong pollution (12/2/2019–3/12/2020).

[Figure]

**Figure 6.** OPS performance during the period of mild pollution (3/13/2020–5/4/2020).

[Figure]

**Figure 8.** Long-term comparisons of MAQS sensor with BAM operated by US EPA at the nearby location: hourly, daily, monthly average values and comparison of hourly and daily average values of two MAQS sensors: first one (MAQS) at our main facility and second one (MAQS-FEE) at Faculty of Electrical Engineering in the immediate vicinity of BAM-1020.

190 **- The Alphasense is now on version OPC-N3 and it is hard to find information on earlier version idem on the PMS5003, they are now at PMS7003. Could you comment if you expect the observations here to be transposable otherwise they are useless.**

Response: OPC-N3 is a successor to OPC-N2. By analyzing the specifications of both, we expect that these results are applicable to OPC-N3 as well. Major differences between OPC-N3 and OPC-N2 are internal temperature and humidity sensor 195 in OPC-N3 and slightly lower detection limit, 350 nm (N3) instedad of 380 nm (N2). While these are useful features, they certainly don't make OPC-N2 obsolete.

Plantower PMS7003 is a miniaturized version of PMS5003. The only advantage of PMS7003 is its smaller size. Because of that PMS7003 is preferred if the size is critical, for example the AirBeam project (Mukherjee, 2017, Sensors). But for our research PMS5003 is more appropriate, because it is more rugged and it has larger intake fan. PurpleAir (Tryner, 2020, AE) is good example of integration of PMS5003 into the network of sensors. We have many laboratory test results of PMS7003 and PMS5003 and conclusion is clear: these sensors give the same results. Please have a look at the results of one of our tests, where we compare these two sensors. Burning chamber with incense scents as the source of PM was used.

[Figure]

**Comparison of PMS7003 and PMS5003**

**- The description of the PMS device seems very speculative? This is very weird when a simple google gives clear descriptions of the device ( https://www.aqmd.gov/docs/default-source/aq-spec/resources-page/plantowerpms5003-manual_v2-3.pdf )**

Response: we have that file, which contains specifications of PMS5003. However, some important information about the sensor is not published there. For example, there are two modes (SM and AE) and the Manufacturer did not publish what is the difference. But we agree that our description of PMS5003 is not good, and we have changed it.

 It uses red semiconductor laser, photodetector at $90^0$ scattering angle (Kuula et al., 2020) and 32-bit processor (Cypress CY8C4245, 48 MHz). According to (Tanzer et al., 2019) PMS5003 is a nephelometer, not the particle counter.

**- The discussion of RH impacts is very cursory and given how big the issue is, it would be important to see how results of the sensors agree or disagree as a function of RH. Here it would be critical to discuss that the gravimetric measurements are done at a given RH but how does this RH compare to the sensor measurements.**

Response: according to requirements of the standard EN 12341:2014, all filters were conditioned in our gravimetric laboratory at relative humidity between 45% and 50%, and temperature between 19 and 21 $^0$C. Without dryers on all devices, we can not control these parameters outside. However, this campaign provided us with enough measurements to draw some important conclusions about humidity influence on measurements from OPS. This text is added in section 2:

According to requirements of the standard, all filters were conditioned at relative humidity between 45% and 50%, and temperature between 19 and 21 $^0$C.

This text is added in section 3.3:

Figure 8 shows the long-term (13.5 months) comparison of MAQS and BAM-1020 with time resolution of 1 hour, together with measured values of ambient air humidity. By averaging all this data we can estimate the influence of humidity on the MAQS sensor: if we sort the measurements by humidity, subset of points where humidity is below 50% has average bias of

14.3%, for humidity range 50%-70%, bias is 16.5%; for humidity range 70%-85% bias is 31.6% and for humidity rang 85%-
100% bias is 37.3%. If we subtract bias of least humidity subset from bias of highest humidity subset, we can estimate that humidity influence adds up to 23% on PM2.5 readings from MAQS sensor, which is similar result to the analysis of humidity influence on our 11-D with dryer installed. While this influence can not be neglected, it is still relatively modest. Reason for this is the composition of particles, where we have mostly fine particles below 300 nm, for which hygroscopic growth is less pronounced (Kosmopoulos 2020).

**Can you comment that you are running the sensors close to their technical specs (95% RH) also at times you actually do run the Grimm D11 outside of specs as the Grimm specs say temperatures above 4 degC (although you also seem to heat the inlet) this is not very clear.**

Response: primary aim of this research is the evaluation of instruments in realistic scenario. That includes wide range of all operating parameters. During the campaign we followed all recommendations of manufacturers of devices, especially Grimm, to ensure that instruments are running normally. Rigorous quality assurance procedure was used. All measurements below LLOD, outside of the specifications and with error or warning codes in the logs were discarded.

**The introduction needs serious revision. Particulate matter and aerosol is not the same thing. Please eliminate all discussion of aerosol as aerosol is the particles and the gases**

Response: we accept this. Instead of "aerosol", "particulate matter" or other appropriate term will be used consistently through the entire manuscript.

**The introduction is very narrowly focused and does not discuss things like the use of TEOM in networks. Also some of the statements should clearly be supported by references**

Response: we agree that network of TEOMs is an interesting topic, but it is not related to this work. TEOM is not an optical scattering device.

**You are very non quantitative and non rigorous in the text and very imprecise., This needs substantial improvement. E.g. L81: what is meant by extremely high? L85 you know exactly what your lower size limit is, so please state it, L 166 what is mean by a "good" correlation?**

Response: L81 "extremely high" refers to measured value of PM2.5 concentration up to 504.9 $\mu$g/m$^3$. That is extremely high concentration of $PM_{2.5}$. For example, in US AQI categorization, values of $PM_{2.5}$ over 500 $\mu$g/m$^3$ are beyond air quality index scale. We accept objection about lower size limit in L88, so instead

we have

It can detect particles with diameters from 10 nm up to 1 $\mu$m.

$R^2$ coefficients from 0.90 to 0.99 represent very good correlation in this context (comparison of optical devices to the reference gravimetric method).

**The statistical discussion totally lack rigor. L 262 "the correlations for hourly, daily and monthly average values of PM2.5 are 0.919, 0.980 and 0.998, respectively" what does this mean? Followed by "with absolute values overestimated by 20% on average " how was this obtained? Where is the data? This is not obvious from Fig 8 at all?**

Response: Fig 8 is redrawn completely, and now you can see these average bias values on the figure as a function of air humidity. The underlying datasets for this publication are available at

https://doi.org/10.5281/zenodo.3897379

Here are the text changes:

Based on 13.5 months of continuous comparison of MAQS and BAM-1020, hourly average values give $R^2$ coefficient 0.919 and MAE 16.7 $\mu$g/m$^3$. Daily average values produce $R^2$ coefficient 0.980 and MAE 12.2 $\mu$g/m$^3$, while the monthly average values give $R^2 = 0.998$ and MAE $= 11.4$ $\mu$g/m$^3$ (Figure 8).

**- the abstract should not read like an experiential section with study dates etc. These details should not go there instead it should contain quantitative results form the paper.**

[revised manuscript text omitted]

---

## Author Response (AR2)

**Response to the Associate Editor**

We would like to thank Mr. Pierre Herckes for the thoughtful examination of our manuscript! Please find the minor corrections of our manuscript.

**Response to Anonymous Referee #2**

We would like to thank the reviewer for the thoughtful and constructive examination of our paper! Please find below our responses to each comment individually and please note that:

– **Blue bold font** represents comments of the Referee,

– Black regular font represents the response to each referee's comment,

–  represents removed text from the manuscript according to referee's comment,

– Red font represents added text in the manuscript according to referee's comments.

**The authors have made significant improvements to their manuscript. With the popularity of the devices evaluated and the wide range of pollutant concentrations experienced I think the additions make it an excellent addition the field. I just request clarification on the response to one previous comment. Line 251: "Reason for this is the composition of particles, where we have mostly fine particles below 300 nm, for which hygroscopic growth is less pronounced (Kosmopoulos 2020)." I don't understand what you mean by this and I didn't find the answer in a quick skim of Kosmopoulos. Are you saying that particles less than 300 nm grow less than larger particles or something about the chemical composition of the particles? If the detection limits of the sensors are also 300 nm how does that affect this?**

Response: thank you very much for pointing this out. It's a mistake. Here is the correct formulation:

Line 251:

Possible reason is the chemical composition of PM without too much hygroscopic components, but that requires a different type of analysis. Relatively modest humidity influence on PMS5003 was also reported in (Jayaratne et al., 2018) and surprisingly low influence is reported by (Kosmopoulos et al., 2020).

**Line 286: I believe this should be 3/27/2020 not 2000**

Response: yes, the correct date is 3/27/2020.

Line 286:  3/27/2020

**Figure 11: This looks great! Very interesting, thanks for adding this helpful discussion.**

Response: thank you for your suggestion to do that.

**Additional minor changes in the revised manuscript:**

Line 8:  particulate matter

Line 62:

Line 84:  and high bias

Line 93:  10

Line 128:  Figure 3

Line 248:  ,

Line 255:  Figure 9

Line 258:  Figure 10

Line 258:  figures 9 and 10

Line 264:  Figure 10

Line 271:  and can't detect

Line 326:  $PM_{2.5}$

Line 327:

[revised manuscript text omitted]